# The Non-Equilibrium Thermodynamics of Natural Selection: From Molecules to the Biosphere

**DOI:** 10.3390/e25071059

**Published:** 2023-07-13

**Authors:** Karo Michaelian

**Affiliations:** Department of Nuclear Physics and Application of Radiation, Instituto de Física, Universidad Nacional Autónoma de México, Circuito Interior de la Investigación Científica, Ciudad Universitaria, Mexico City C.P. 04510, Mexico; karo@fisica.unam.mx

**Keywords:** origin of life, evolution, disspative structuring, prebiotic chemistry, abiogenisis, adenine, biosphere, natural selection, 87.23.Kg, 87.15.-v, 05.70.Ln, 92-10, 92D15, 92C05, 92C15, 92C40, 92C45, 80Axx, 82Cxx, 82C35

## Abstract

Evolutionary theory suggests that the origin, persistence, and evolution of biology is driven by the “natural selection” of characteristics improving the differential reproductive success of the organism in the given environment. The theory, however, lacks physical foundation, and, therefore, at best, can only be considered a heuristic narrative, of some utility for assimilating the biological and paleontological data at the level of the organism. On deeper analysis, it becomes apparent that this narrative is plagued with problems and paradoxes. Alternatively, non-equilibrium thermodynamic theory, derived from physical law, provides a physical foundation for describing material interaction with its environment at all scales. Here we describe a “natural thermodynamic selection” of characteristics of structures (or processes), based stochastically on increases in the global rate of dissipation of the prevailing solar spectrum. Different mechanisms of thermodynamic selection are delineated for the different biotic-abiotic levels, from the molecular level at the origin of life, up to the level of the present biosphere with non-linear coupling of biotic and abiotic processes. At the levels of the organism and the biosphere, the non-equilibrium thermodynamic description of evolution resembles, respectively, the Darwinian and Gaia descriptions, although the underlying mechanisms and the objective function of selection are fundamentally very different.

## 1. Introduction

The “Modern Synthesis”, combining Mendelian genetic theory with Darwinian evolutionary theory incorporating reproduction with mutation, random genetic drift, gene flow, and natural selection, provides a conceptual framework for assimilation of much of the biological and paleontological empirical data concerning biological evolution at the level of the organism. However, fierce criticism (although not always scientific) has been leveled at the theory. The first serious scientific challenge concerned the theory’s lack of physical foundation. In the 1957 book “The Poverty of Historicism”, philosopher of science Karl Popper argued, “There are neither laws of succession nor laws of evolution.” [1]. Popper conceded that there were historical trends in nature, but argued that trends cannot be characterized as universal generalizations and, therefore, are not laws [2].

In “The Myth of the Framework” (1994) [3] Popper sharpened his criticism of evolutionary theory, arguing that if the common statistical definition of fitness (differential reproductive success of individuals within populations) were accepted “then the concept of the survival of the fittest becomes tautological, and irrefutable”. Since Popper insisted that a “scientific” theory be refutable [4], this left him with no choice other than to refer to Darwinian theory as a non-scientific “metaphysical research program”.

Contrary to what is often claimed in the literature, Popper did not recant on this position latter in life [5]. He did, however, concede that Darwinian theory could, nevertheless, be useful as “a possible framework for testable scientific theories” [6]. To Popper, Darwinian theory was not pseudoscience, but simply lacked a foundation based on fundamental empirical law which would make the theory testable [5]. The lack of physical foundation of evolutionary theory has rarely been addressed since Popper’s criticism. This appears to be due, in part, to the inherent complexities of the issues, but also to a zealous suppression of any criticism, probably as an over-reaction to the aggressive non-scientific skepticism of the theory expressed by religious adherents.

The lack of foundation, however, manifests itself in the many unresolved problems and paradoxes. Examples include; the theory’s inability to shed light on the origin of life, the tautology in founding a “natural selection” on differential reproductive success, the problem of separating the environment from the organism (e.g., whether the environment is selecting the organism or vice-versa, and sexual selection wherein the selective environment experienced by one sex is created by properties of the other), explaining the paleontological evidence for punctuated equilibrium (long periods of stasis interrupted by rapid change [7]), the “paradox of the first tier” (failure to find expected “progress” among competing organisms, e.g., simple prokaryotes never surrendering their vast kingdom to more “evolved” organisms [7]), apparently “directed” mechanisms of change distinct from random mutation (e.g., horizontal gene transfer — viral vectors, Lamarckian evolution — epigenetics, genetic drift—trans-splicing, societal and cultural evolution), selection apparently operating at different levels from the gene to the biosphere (e.g., Dawkin’s “selfish gene” [8]), and the concept of “reproductive success” losing interpretation at the higher levels (e.g., the ecosystem and biosphere) known as the “paradox of the evolution of a system of population one” [9].

Although these problems have not been ignored, they have usually been addressed only from “within the box”, i.e., through judicial extensions attempting to keep the basic tenants of traditional evolutionary theory intact. This, however, has resulted in little further elucidation and has often led to new paradoxes. For example, Richard Dawkins in his book “The Selfish Gene” [8] questions the level at which natural selection was really operating, suggesting that it might be at the gene level, demoting the organism itself to the gene’s “survival machine” [8]. Others have considered natural selection operating at the species level (macroevolution) [10]. James Lovelock emphasized the empirical data pointing to mutual evolution of the entire biosphere, and argued for a collective, rather than individual, fitness function, leading to biological control of planetary processes assuring “suitability for all life”. In his most enthusiastic presentation, Lovelock considered the Earth itself as a homeostatic living organism and gave it the name Gaia, after the Greek goddess of Earth [11]. Steven Gould criticized this “strong Gaia hypothesis”, suggesting it was only a metaphor, rather than a detailed mechanistic theory [7]. Neo-Darwinism itself, however, has not escaped this same criticism [12]. Gould did, however, realize that selection was occurring simultaneously on many levels (living system hierarchies) and tiers (distinct time periods relevant to ecological processes, e.g., speciation, punctuated equilibrium, or extinctions) but his inability to find a satisfactory resolution of these issues from within the traditional paradigm led him to the same conclusion as Popper, that evolutionary theory was, at the very least, incomplete [7].

Perhaps the most conspicuous indication of the incompleteness of evolutionary theory, however, is that it sheds no light on the origin of life. Here too, attempts have been made to stretch the fabric of traditional evolutionary theory to cover selection of individual molecules, for example, based on chemical stability, or on the ability of molecules to sequester molecular precursors through chemical affinity. Neither of these two lines of research have, however, proved fruitful.

Alternatively, non-equilibrium thermodynamic theory in the non-linear regime, as developed by Onsager, Machlup, Prigogine, Nicolis, Glansdorff and others [13,14,15], provides a physical/chemical description of the complex dynamics of material interacting with its environment. Under this framework, processes, known as *dissipative structures* (flows) [16,17] arise “spontaneously” under an imposed external thermodynamic potential (force) to dissipate this potential. Under given external conditions (external forces), multiple, locally stable *stationary states* exist for non-linear systems and this means that, under perturbation, the material system evolves over these states, directed by both stochastic fluctuations and deterministic thermodynamic law derived from the second law and the conservation laws (related to fundamental symmetries of nature).

The advantage of the thermodynamic framework over traditional evolutionary theory is that it is based on established fundamental law; the conservation laws, the second law of thermodynamics, and the existence of various solutions to, and instabilities in, non-linear systems, and furthermore applies simultaneously to all hierarchal levels, from the fundamental molecules at origin of life, to the present biosphere. Stubborn problems and paradoxes of traditional evolutionary theory find resolution under this non-equilibrium thermodynamic framework, and a physical-chemical explanation of the origin, persistence, and evolution of life can be provided.

Analysis of biological systems from the perspective of irreversible thermodynamic theory has led to an understanding of many of the extraordinary characteristics of biology and its interaction with its environment, without requiring the ad hoc assumption of an inherent “vitality” of living material plaguing the biological perspective [14,17,18,19,20,21,22,23,24,25,26].

## 2. A Non-Equilibrium Thermodynamic Framework

Dynamics involving material or energy (i.e., flows or processes) are driven by what are known as *generalized thermodynamic potentials*. Examples include the electric potential which promotes a flow of charged material (a current), temperature potentials which promote a flow of energy (heat flow), concentration potentials which promote a flow of matter (diffusion), chemical and photochemical potentials which promote the process of molecular transformation (chemical and photochemical reactions), etc. The gradients of these potentials with respect to space or concentrations (e.g., temperature and concentration spatial gradients, or chemical and photochemical affinities, respectively) divided by the local temperature, are known as *generalized thermodynamic forces* because the size of this quantity determines the strength of the corresponding flow (e.g., heat or mass flow, or reaction rate), which are known as *generalized thermodynamic flows*.

When thermodynamic forces are sufficiently large, the relation between force and flow is not necessarily linear because the production of internal flows can lead to new internal forces, and so on, giving rise to a set of coupled processes with feed-back and thus non-linear behavior. It is precisely this non-linearity between forces and flows that leads to many solutions and thus to the exuberant diversity of dynamics seen in the interaction of material with its environment, especially for carbon based material when subjected to a strong photochemical potential (i.e., life). In the following sections we explore the possibility that the origin of life was a particular scenario of the dissipative structuring of carbon based material under the UV-C light potential of the Archean, and that the subsequent evolution of life corresponded to the continuation of this dissipative structuring into the higher intensity visible wavelengths and over the entire surface of Earth, so as to increase global photon dissipation in accordance with non-equilibrium thermodynamic directives.

### 2.1. Origin of the Second Law and Irreversible Thermodynamic Theory

All generalized thermodynamic forces are ultimately manifestations of the second law of thermodynamics and the symmetries of nature. The second law captures the empirical fact that the conserved quantities of nature (e.g., energy, momentum, angular momentum, charge, etc.) “spontaneously” become, over time, globally (system plus environment) distributed over ever more microscopic degrees of freedom, including vibrational, rotational, translational, and electronic (and nuclear at very high temperature) of the molecules or atoms of the material, and that this evolution is irreversible in time. This is considered to be a fundamental law of nature, valid macroscopically rather than microscopically.

Why the second law of thermodynamics (irreversibility) is part of nature, but not envisaged in our descriptions of both classical and quantum mechanics, has been debated for more than 150 years. A favored perspective is that it arises from quantum decoherence [27,28] related to the practical impossibility of completely isolating any system from its environment [29]. Quantum fluctuations of the environment have relevance to this, and, together with measurement effects, give rise to the Heisenberg uncertainty relation [30]. We will see in Section 4 that the Heisenberg uncertainty relation implies a finite minimum value for entropy production within a finite time interval Δt for any macroscopic process, or in other words, quantum fluctuations are the source of irreversibility.

Others believe that the second law arises from the non-linearity in nature, or the expansion of the universe. Still others (e.g., Einstein) believed that irreversibility is only an illusion, since time itself is an illusion. Another proposal by Ilya Prigogine [27] is the realization that individual and integrable trajectories of elementary particles, for which classical physics, quantum mechanics, and equilibrium thermodynamics were developed, is only an idealization. Instead, real systems, even such simple systems as three bodies interacting gravitationally (the three-body problem), are non-integrable because there exists persistent interaction between particles, or, non-local (in space and time) resonances giving rise to infinities on integration. In this case, a probabilistic, rather than deterministic, representation of the fundamental laws is required to properly describe the dynamics, and this necessitates the introduction of the second law of thermodynamics, with its signature of global entropy increase over time (i.e., irreversibility). Most of these explanations are probably related.

Irrespective of the controversy, most scientists are content to consider the second law of thermodynamics as fundamental since no macroscopic violations have yet been observed. It is the second law that we also adopt here as being fundamental to an understanding of evolution of material interacting with its environment from a physical-chemical perspective. Furthermore, we also accept as valid the results of Classical Irreversible Thermodynamic (CIT) theory developed by Onsager, Prigogine, and co-workers [13,14,15] which follow from the second law and the conservation laws under specific environmental conditions. CIT theory in the non-linear regime establishes the existence of multiple, locally stable, dissipative stationary states, consisting of processes (flows) which can break space and time symmetry (dissipative structuring), for a system interacting with its environment. CIT theory has been empirically validated for situations in which a macroscopic *local thermodynamic equilibrium* in space and time can be established, a restriction observed for most terrestrial, sufficiently dense, material systems [14], including “fast” systems involving, for example, photon-induced quantum electronic excitations [31].

### 2.2. Multiple Stationary States

For material systems under an imposed constant generalized thermodynamic potential from the environment, the material in the system will develop the corresponding internal flow (of mass, energy, charge, reaction rate, etc.) and induced secondary internal forces, but will eventually come to what is known as a *thermodynamic stationary state* in which the generalized thermodynamic flows and forces, and the entropy production, arrive at constant values at any given point within the system (although being potentially different throughout the system—see Figure 1). The total entropy of the material within the system could either increase or decrease [14], while the entropy of the system plus its environment necessarily increases, in observance of the second law of thermodynamics.

For systems in which flows are non-linearly related to the forces, there can be more than one stationary state available to the system for the same boundary and initial conditions because non-linear equations have multiple solutions. Such stationary states are, therefore, only locally stable and large enough perturbations, particularly near *critical points*, can cause the system to evolve from one stationary state to another. For particular stationary states with non-linear positive feedback (for example, an auto-catalytic chemical reaction set dissipating an imposed chemical potential), such states can be imagined to be more stable locally than other allowed states (for example, a non-catalytic set of reactions) because statistical microscopic fluctuations would be less likely to alter the reaction route since the route with positive feedback (catalyzed) dissipates a greater amount of the free energy entering the system (it is the availability of this free energy which could potentially alter the route). In other words, the basin of attraction in a generalized phase space will be larger for the auto-catalyzed route (stationary state) and there will be a greater statistical probability of evolution, on stochastic perturbation, towards those stationary states with greater dissipation of the imposed potential. This will be demonstrated explicitly in Section 4 with an example of the photochemical dissipative structuring of adenine, one of the fundamental molecules of life (those molecules found in all three domains of life), from common precursor molecules in water solution under an imposed constant UV-C photochemical potential.

### 2.3. The Dynamics of Material-Environment Interaction

In summary, the elements of non-linear CIT theory which describe the dynamics of a material system interacting with its environment are:The existence of at least one applied external generalized thermodynamic potential defining the environment, and the gradients of these (divided by the local temperature), which are the applied thermodynamic forces.The spontaneous generation of internal generalized thermodynamic flows derived from the applied external generalized forces and the new internal forces that these flows themselves may generate.The existence (in the asymptotic time limit) of various sets of these internal (to the system) forces and flows for non-linear systems for the same initial and boundary conditions, (i.e., multiple, locally stable, dissipative structures, or processes, at stationary states), which can have different rates of dissipation of the applied external potential (entropy production).External stochastic perturbations, or internal fluctuations, which could cause the non-linear system to leave the local attraction basin of one stationary state and evolve to another, particularly near a critical point (along borders of the attraction basins of different stationary states).The statistical (non-deterministic) tendency for evolution on perturbation to stationary states (dissipative structures) of greater dissipation (entropy production) having positive feed-back, giving them a larger “attraction basin” in a generalized phase space.

This list of elements characterizing non-linear CIT theory, based only the conservation laws and the second law of thermodynamics, and assuming the establishment of local equilibrium, is sufficient to provide a physical and chemical explanation of the origin and evolution of life [21,23,25,32,33]. Using these elements of the theory, we will now show how the macroscopic concentration profiles of UV-C pigments (which we claim became the fundamental molecules of life; i.e., nucleic acids, amino acids, sugars, fatty acids, metabolites and cofactors, and pigments) arise through dissipative structuring from concentrations of simpler and common precursor molecules under UV-C light, in order to dissipate this light efficiently into heat. In order to understand this process at a microscopic level, and therefore avoid the usual criticism of metaphoric speakeasy, it is necessary to specify in detail the mechanisms involved. First, however, it is important to review the basics of organic photochemistry.

## 3. The Photochemistry of Organic Molecules

The fact that a sun deity usually reigned supreme in past cultures suggests that humans have, consciously or subconsciously, always understood that the sun was at the foundation of life. This commemorated intuition turns out to be a good starting point for building a modern framework from within which to understand life. The “thermodynamic dissipation theory for the origin and evolution of life” considers life as a non-equilibrium thermodynamic process arising out of the imposed force or potential, the incident solar light flux at Earth’s surface, giving rise to the dissipative structuring of pigments (the fundamental molecules of life) from common carbon based precursor molecules in water solvent.

Analogous to heat flow, or a convection cell, arising “spontaneously” under a temperature gradient to dissipate this gradient, the thermodynamic dissipation theory for the origin of life suggest that the fundamental molecules of life (nucleic acids, amino acids, sugars, fatty acids, cofactors, and their complexes) arose “spontaneously” as UV-C pigments, dissipatively structured on the ocean surface from common precursor molecules such as hydrogen cyanide HCN, cyanogen (CN)_2_, carbon dioxide CO_2_ and water under the soft UV-C photon flux (between approximately 205 and 285 nm, Figure 2) arriving at Earth’s surface throughout the Archean [21,23,25,32,33]. This wavelength region has sufficient energy per photon to transform carbon covalent bonds (single and double), but generally not enough energy to sufficiently ionize or fragment these molecules.

The best geochemical evidence presently available suggests that this light would have been present on Earth’s surface from before the origin of life (at ∼3.9 Ga) and for at least 1000 (perhaps even for 1500 [34]) million years until organisms evolved oxygenic photosynthesis which saturated available oxygen sinks [34,35,37], leading to a protective ozone layer in the stratosphere. Additionally, cyanobacteria (and today’s plants) emit volatile organic compounds (VOCs) which, together with atmospheric oxides of nitrogen, are the precursors for photochemical production of ozone in the troposphere [38,39]. Oxygen and ozone are thus considered here as biology-procured pigments, today dissipating the UV-C region in the upper atmosphere, thereby allowing the present dissipative structuring of the more delicate (i.e., prone to UV-C disassociation) complex biosynthetic pathways producing visible pigments at Earth’s surface using multiple visible photons of lower energy but of much higher intensity.

A comprehensive description of the thermodynamic dissipation theory for the origin of life, including dissipative molecular structuring, the origin of homochirality, enzymeless nucleic acid replication, early information encoding, and protocell formation, has been given elsewhere [21,23,25,32,33,40,41,42,43,44]. The purpose of this paper is to analyze in detail the mechanism of thermodynamic selection, operating at the molecular level at the origin of life, and subsequently at higher biotic levels, up to the present biosphere. Life’s evolutionary history will be shown to be driven by a natural thermodynamic selection based on efficacy of photon dissipation of the biosphere. The paleontological evidence indeed indicates greater photon dissipative pigment concentration profiles over life’s history [36,45].

The following subsection describes how UV-C light interacting with carbon-based precursor molecules structures these into complex UV-C pigments (the fundamental molecules of life, Figure 2) with conical intersections to internal conversion, providing efficient photon dissipation.

### 3.1. Absorption of Light and Its Dissipation through Internal Conversion

The absorption of UV-vis light in organic molecules is a phenomenon involving collective electron excitation requiring conjugation of carbon bonds (alternation of double and single carbon-carbon covalent bonds—Figure 3). Conjugation, implies removing hydrogen ions from saturated hydrocarbons to form carbon-carbon double bonds (C = C) thereby freeing electrons from their atomic orbitals. This provides a new set of collective electronic excited states involving a number of atomic nuclei now sharing their valence electrons. Atomic electronic orbitals thus become molecular orbitals which can be either bonding or anti-bonding between atoms. The greater the conjugation number, the greater the wavelength of peak absorption of the molecule (Figure 3). The energy difference between the ground state and first excited state for doubly and triply conjugated carbon based molecules corresponds to the range of energies of the soft UV-C photon region arriving at Earth’s surface during the Archean (Figure 2).

Because of its thermodynamic importance (to dissipation), the facility to form, and stability of, these UV-C light absorbing conjugations in carbon based molecules are probably the principle reasons that Earth’s life is based on carbon. The other chemical properties of carbon-based molecules [46] are, therefore, likely only of secondary importance to life. For example, elements with similar outer-shell electronic structure, like silicon, also support 4 covalent bonds, but silicon molecules only rarely present conjugation because Si-O bonds are significantly stronger than Si-Si bonds. In fact, double Si=Si bonds are unstable in aqueous solution [46]. Under the particular UV-C light environment at Earth’s late Hadean and early Archean ocean surface, it is thus probable that only carbon based molecules, and complexes of these, could be dissipatively structured into stable and efficient UV-C pigments.

Empirical evidence supports our UV-C dissipative structuring conjecture for the origin of the fundamental molecules of life:The free energy available in UV light of wavelength less than 300 nm arriving at Earth’s surface today is more than 1000 times greater than that of all other non-photon energy sources combined [47], and this may have been even greater at the beginning of the Archean because of the lack of an ozone layer.The wavelength of maximum absorption of many of the fundamental molecules coincide with the predicted window in the Archean atmosphere (Figure 2).The wavelength of maximum absorption of the fundamental molecules can be tuned very simply by a protonation or deprotonation event, decreasing or increasing, respectively, this wavelength by about 30 nm (Figure 3). This allows carbon based dissipative structures to easily evolve towards dissipation of higher intensity light at longer wavelengths and to adapt to a changing surface solar spectrum.Many of the fundamental molecules of life are endowed with peaked conical intersections [23,48,49] giving them broad band absorption and high quantum yield for internal conversion, i.e., extremely rapid (sub-picosecond) dissipation of the photon-induced electronic excitation energy into vibrational energy of molecular atomic coordinates, and finally into the surrounding water solvent. It is the photon-induced excitation of an electron into an anti-bonding (e.g., $\pi^{*}$) orbital that weakens the respective bond, decreasing the energy of the excited state upon elongation of the bond, leading to intersection of the excited potential energy surface with the potential energy surface of the ground state (Figure 4).Many photochemical routes from common and simple precursor molecules to the synthesis of nucleic acids [50], amino acids [51], fatty acids [42], sugars [52,53], and other pigments [36] have been identified at these UV-C wavelengths.The rate of photon dissipation within the Archean UV-C window generally increases after each incremental transformation on route to synthesis of the fundamental molecule (Figure 5), a behavior strongly suggestive of dissipative structuring in the non-linear non-equilibrium thermodynamic regime [23,25,42].Even minor transformations (e.g., tautomerizations or methylations) of the fundamental molecules, which often, in fact, endows them with lower Gibb’s free energy, eliminates, or significantly reduces, their extraordinary photon absorption and dissipation properties [54].

These extraordinary photochemical properties of the fundamental molecules of life in the UV-C region would have to be considered as mere coincidences if UV-C light, and light absorbing carbon conjugations, had no connection with the origin of life.

Conical intersection seams on the excited potential energy surface, in multi-dimensional atomic coordinate space, determine the photoisomerization or photoreaction products that can be reached after a photon absorption event. Since conical intersections are located energetically down-hill from the Franck-Condon region (Figure 4), the direction and velocities of approach of the nuclear coordinates to a conical intersection are important in defining the outcome [49]. For example, it is known that for the molecule retinal in rhodopsin, the photoexcited molecule reaches the conical intersection extremely fast (75 femtoseconds) implying that the conical intersection must be peaked (inverted cone-like on the excited state potential energy surface) and, overwhelmingly, only one reaction product is reached, which for the case of retinal, as well as for the fundamental molecules of life, is the original ground state configuration [55]. A more extended seam with different minima can lead to different reaction products [56] such as those intermediate molecules on route to the photochemical synthesis of the nucleobase adenine (Figure 5). In summary, the final products of the photochemical dissipative structuring (the fundamental molecules of life) have a peaked conical intersection to internal conversion allowing them to dissipate the electronic excitation energy to the ground state extremely rapidly and thus become the final and photo-stable product of dissipative structuring in the relevant region of the solar spectrum.

**Figure 4 entropy-25-01059-f004:**
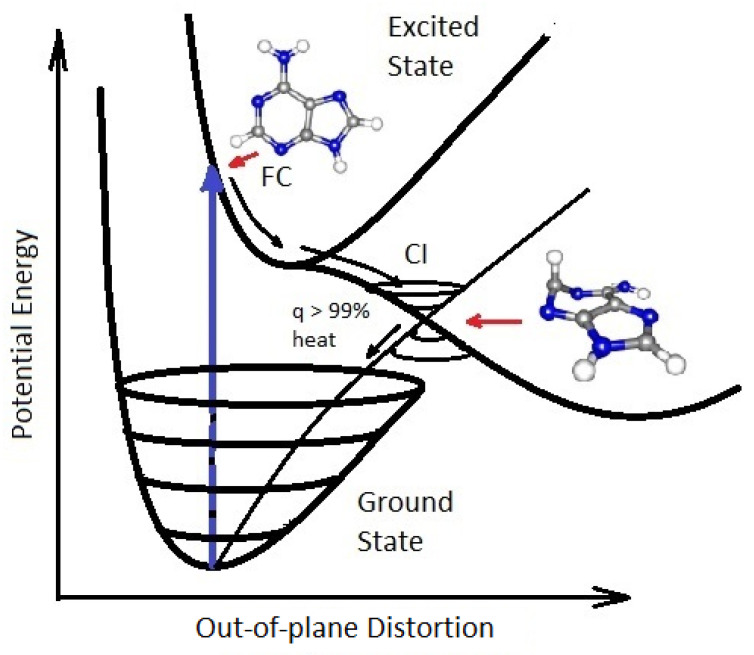
A conical Intersection (CI) for excited adenine showing a degeneracy of the electronic excited state with the electronic ground state after a UV-C photon absorption event (blue arrow) which induces a nuclear coordinate deformation from its original structure in the Franck-Condon (FC) region, either activation of an N9–H stretch or a ring-puckering motion known as *pyramidalization* (shown in the diagram). The most probable deformation depends on the incident photon energy and protonation state. It is this deformation resulting from excitation to an anti-bonding state (e.g., $\pi \to \pi^{*}$) which leads to a lowering of the excited potential energy surface such that it intersects with the electronic ground state. Conical intersections provide rapid (sub-picosecond) dissipation of the electronic excitation energy into vibrational energy (heat). The quantum efficiency, *q*, for this dissipative route is very large (>99%) for many of the fundamental molecules of life, making them photochemically stable and, more importantly for our theory, very efficient at UV-C photon dissipation. Another common form of coordinate transformation mediated through conical intersections are proton transfers within the molecule or with the solvent environment and this may have relevance to a photon-induced denaturing of RNA and DNA (see below). Based on data from Andrew Orr-Ewing [57], Roberts et al. [58], Kleinermanns et al. [59], and Barbatti et al. [60]). Reprinted with permission from Ref. [25].

It has been a recurrent theme in the literature that the rapid (sub-picosecond) deexcitation of the excited nucleobases due to their conical intersections had evolutionary utility in providing stability under the high flux of UV photons that penetrated the Archean atmosphere [35,61] since a peaked conical intersection reduces the lifetime of the excited state to such a degree that further chemical transformations are improbable. Although this is true, photo-stability is never perfect for photon absorbing molecules, and photochemical reactions under UV-C light still occur for the fundamental molecules of life, particularly after excitation to the long-lived triplet state (e.g., cyclobutane pyrimidine dimers, 6-4, and other photo-products, in RNA and DNA [62]). An apparently more optimal and simpler solution for avoiding radiation damage with its concomitant degradation in biological function, therefore, would have been the synthesis of molecules transparent to, or reflective to, the offending UV light (e.g., saturated hydrocarbons or molecules made from silicon atoms lacking the possibility for conjugation needed for photon absorption). From the perspective of the thermodynamic dissipation theory for the origin of life, however, a large antenna for maximum UV-C photon absorption and a peaked conical intersection for its rapid dissipation into heat are, in fact, precisely the thermodynamic “design goals” of dissipative structuring.

**Figure 5 entropy-25-01059-f005:**
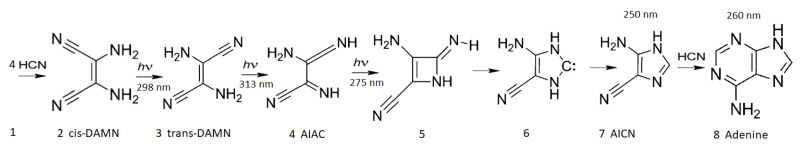
The photochemical dissipative synthesis of adenine from 5 molecules of hydrogen cyanide (HCN) in water, as discovered by Ferris and Orgel (1966) [50,63]. Four molecules of HCN (1) are transformed into the smallest stable oligomer (tetramer) of HCN, known as cis-2,3-diaminomaleonitrile (cis-DAMN) (2), which, under a constant UV-C photon flux isomerizes into trans-DAMN (3) (also known as diaminofumaronitrile, DAFN) which can be converted, on absorbing two more UV-C photons, into an imidazole intermediate, 4-amino-1H-imidazole-5-carbonitrile (AICN) (7). Hot ground state thermal reactions with another HCN molecule or its hydrolysis product formamide (or ammonium formate) leads to the purine adenine (8). This is a microscopic dissipative structuring process which ends in adenine [23,25], a UV-C pigment with a large molar extinction coefficient at the maximum intensity of the UV-C Archean solar spectrum (260 nm) and a peaked conical intersection which promotes the dissipation of photons at these wavelengths (Figure 2). Note that a non-photochemical route to adenine from HCN using ammonia and water catalyzed steps at higher temperatures has been suggested in reference [64]. Based on data from Ref. [50].

## 4. Natural Thermodynamic Selection at the Origin of Life

Given the probable existence of common carbon based molecular precursors during the Archean, such as hydrogen cyanide (HCN), cyanogen (CN)_2_, and carbon dioxide (CO_2_) [25,33], in a water solvent, under soft UV-C light between approximately 205 and 285 nm (Figure 2), photochemical and chemical (thermal) reactions will occur “spontaneously” to dissipate the respective photochemical and chemical potentials. Product molecules resulting from these reactions that absorb less well the incident soft UV-C spectrum will become less susceptible to further photochemical transformation, although still subject to further chemical reactions. Chemical reactions at Earth surface temperatures, however, result in much less product variety as compared to photochemical reactions in which the deposited photon energy is sufficient to overcome high activation barriers [25]. On the other hand, those product molecules that become progressively more photon absorptive in the soft UV-C region will tend to continue to transform until the quantum efficiency for dissipation of the photon-induced excited state energy directly to the ground state (internal conversion) out-competes the quantum efficiencies for further transformation (i.e., they develop conical intersections to rapid internal conversion—Figure 4). This makes the product molecules photochemically stable, but, more importantly from our perspective of the thermodynamic dissipation theory for the origin of life, they become strongly photon dissipative.

This mechanistic description of microscopic dissipative structuring of carbon based pigments under a soft UV-C light flux is analogous to the mechanistic description of the dissipative structuring of convection cells in a liquid held under gravity and a temperature gradient. In the latter case, at a particular “critical point” in the applied temperature gradient, buoyant forces acting on the less dense hot fluid overcome the opposing force of gravity and the liquid viscous forces, resulting in the “spontaneous” appearance of convection cells under the temperature gradient, which act to increase dissipation of this temperature gradient in comparison to conduction alone. In the pigment dissipative structuring process, for sufficient incident photon intensities at wavelengths shorter than some critical energy, allowing excitation to the first collective electronic excited state of a particular precursor molecule, the system will gradually transform from the original poorly absorptive concentration profiles of the precursor molecules towards the strongly absorbing profiles of the product molecules, increasing the systems photon dissipation efficacy of the imposed external photon potential [23].

Figure 6 describes in detail how the initial precursor molecular concentration profile will gradually transform (evolve) towards a concentration profile with greater amounts of adenine, giving greater photon dissipative efficacy under the impressed soft UV-C photon spectrum of the Archean. Even though the product molecules can (but not necessarily do) have a lower Gibb’s free energy than that of the precursor molecules from which they evolved, in thermal systems (chemical reactions) the evolution to the lower free energy state is not spontaneous if there are large energy barriers between configurations. However, coupling of the reactions to the impressed UV-C photon potential (photochemical reactions), allows the transformation to proceed over the barriers, and even to higher Gibb’s free energy configurations, at a rate dependent on, (i) photon intensities at the different wavelengths, I(λ), (ii) the absorption as a function of wavelength A(λ), and (iii) on the widths of the phase-space paths leading to the particular conical intersection on the electronic excited state potential energy surface (Section 3) (i.e., the quantum efficiencies $q_{i}^{j}$) leading to the molecular transformation i→j. Backward transformations $q_{j}^{i}$, or transformations to other possible products (e.g., $q_{i}^{k}$), under the UV light are less probable if the quantum efficiencies are smaller (smaller phase-space path on the excited potential energy surface) as compared to its quantum efficiency for internal conversion to the ground state $q_{j}^{IC}$.

The condition for evolution of the macroscopic molecular concentration profile in the forward direction (Figure 5 and Figure 6), towards an increasing concentration of molecule *j* (assuming, for simplicity, only two possible molecular configurations, *i* and *j*) is thus;
(1)$$\int_{0}^{\infty }I\left(\lambda \right)\cdot A_{i}\left(\lambda \right)\cdot \frac{q_{i}^{j}\left(\lambda \right)}{\left(q_{i}^{IC}\left(\lambda \right)+q_{i}^{j}\left(\lambda \right)\right)}d\lambda >\int_{0}^{\infty }I\left(\lambda \right)\cdot A_{j}\left(\lambda \right)\cdot \frac{q_{j}^{i}\left(\lambda \right)}{\left(q_{j}^{IC}\left(\lambda \right)+q_{j}^{i}\left(\lambda \right)\right)}d\lambda$$
where $q_{i}^{IC}\left(\lambda \right)$ is the quantum efficiency for internal conversion to the ground state for molecule *i* at wavelength λ.

Since (for our simplified model of only two molecular configurations, *i* y *j*),
(2)$$q_{i}^{IC}\left(\lambda \right)+q_{i}^{j}\left(\lambda \right)=1\;\mathrm{and}\;q_{j}^{IC}\left(\lambda \right)+q_{j}^{i}\left(\lambda \right)=1,$$ Equation (Equation 1) for the condition for evolution of the macroscopic concentration profile in the forward direction can be written as;
(3)$$R_{i\to j}\equiv \int_{0}^{\infty }I\left(\lambda \right)\left[A_{i}\left(\lambda \right)\cdot q_{i}^{j}\left(\lambda \right)-A_{j}\left(\lambda \right)\cdot q_{j}^{i}\left(\lambda \right)\right]d\lambda >0,$$
where $R_{i\to j}$ is the molar rate of conversion of species *i* to species *j*, $A_{i}\left(\lambda \right)=\epsilon_{i}\left(\lambda \right)c_{i}l$ with $\epsilon_{i}\left(\lambda \right)$ the molar absorption coefficient of species *i*, $c_{i}$ the concentration of species *i*, and *l* the path-length of the light. Using Equations (Equation 2) in Equation (Equation 3), the condition for evolution in the forward direction is,
(4)$$R_{i\to j}=\int_{0}^{\infty }I\left(\lambda \right)\left[A_{i}\left(\lambda \right)\cdot \left(1-q_{i}^{IC}\left(\lambda \right)\right)-A_{j}\left(\lambda \right)\cdot \left(1-q_{j}^{IC}\left(\lambda \right)\right)\right]d\lambda >0.$$

This condition will be satisfied if, (1) molecule *j* absorbs less photons than molecule *i* (i.e., that $\int I\left(\lambda \right)\epsilon_{j}\left(\lambda \right)d\lambda <\int I\left(\lambda \right)\epsilon_{i}\left(\lambda \right)d\lambda$), and/or, (2) the quantum efficiency for dissipation through the conical intersection to the ground state is greater in general (integrated over wavelength) for molecule *j* than it is for molecule *i*, i.e., $\int q_{j}^{IC}\left(\lambda \right)d\lambda >\int q_{i}^{IC}\left(\lambda \right)d\lambda$.

Therefore, the molecular concentration profile could either (1) evolve towards a profile with little absorption of light, or (2) towards a profile strongly absorbing light, but with large quantum efficiency for the rapid dissipation of this light through a conical intersection to the ground state. Assuming, for simplicity for the moment, no other coupled dissipative process, and ignoring small changes in the structural (configurational) entropy of the molecules (compared with the entropy production of UV photon dissipation into heat), evolution of a macroscopic concentration profile of molecules towards a less absorptive profile under a constant light flux would be a violation of the second law of thermodynamics which demands that, in any macroscopic process, the entropy of the system plus environment increases. To avoid a macroscopic violation of the second law, therefore, the evolution must be towards a stronger absorbing concentration profile with a large quantum efficiency for the rapid dissipation of the excitation energy to the ground state through a conical intersection.

The specific balance equations, including empirical rate constants, for the particular case of the photochemical dissipative structuring of adenine (Figure 5) and guanine are given in references [25] and [33], respectively. The flows into and out of the vesicle volume are the flows of photons and precursor organic molecules, the latter flow being dependent on the diffusion constants which depend on the size of the molecule and its dipole moment [25]. Such a photochemical dissipative structuring model leads to increases in the product adenine concentration up to 6 orders of magnitude (from $10^{-10}$ M to $10^{-4}$ M) in only 60 Archean days starting with a realistic ocean microlayer concentration of HCN of $6\times 10^{-5}$ M [25,33].

The product molecule characteristics of strongly absorbing over a greater wavelength region and rapid dissipation, are, in fact, complementary since rapid dexcitation to the ground state implies a large wavelength bandwidth for absorption by the Heisenberg uncertainty relation ΔtΔE≥ℏ/2. Considering a macroscopic thermodynamically open system of organic molecules in water at temperature *T* contained within a volume *V* (e.g., defined by a fatty acid vesicle, Figure 7) under a flux of photons with an amount of energy ΔE absorbed within the volume within a time period Δt, the smallest possible entropy production for such a macroscopic process, carried out at constant volume *V* and particle number *N*, would thus be,
(5)$${\left.\frac{\Delta S}{\Delta t}\right|}_{V,N}={\left.\frac{\Delta E}{T\Delta t}\right|}_{V,N}\geq \frac{\hbar }{2T\cdot {\left(\Delta t\right)}^{2}}$$
indicating that, for any ΔE, short period Δt processes (e.g., rapid internal conversion through a conical intersection) at lower temperatures *T* lead to greater minimal entropy production.

This example shows how the Heisenberg uncertainty principle arises from the second law of thermodynamics (or vice versa), because Equation (Equation 5) indicates that entropy production ΔS/Δt in any finite time Δt process at any finite temperature *T* must be positive definite since the right hand side of Equation (Equation 5) is positive definite. This result, that the Heisenberg uncertainty relation is related to the second law, is something already known [65] but this simple example of molecular thermodynamic selection makes the relation obvious and shows the inherent discreteness of the second law with a finite minimum possible increment in the entropy for any real macroscopic process. Note also that Equation (Equation 5) implies that the minimum entropy production increases quadratically with decreasing Δt (shorter time processes) and is always a positive definite quantity, even for time reversed macroscopic processes (i.e., for Δt<0). A minimal entropy production for any macroscopic process implies irreversibility (see Section 2.1).

As noted above, the Heisenberg uncertainty principle does not apply to one single process of a pair of coupled irreversible processes. For example, if the quantum efficiency for de-excitation through an additional conical intersection to photon-induced ionization and subsequent molecular disassociation is included in Equation (Equation 1), then this would be a coupled irreversible process that could occur macroscopically provided that the final disassociated molecular components have higher entropy than the original precursors. However, given the myriad of photochemical reactions available for carbon based molecules under soft UV-C light (205–285 nm), such a local stationary state, if it were visited by the system would be unstable because of little dissipation (only due to increased molecular configurational entropy and photon reflection into a greater solid angle without changing wavelength, which is always less than UV-C photon dissipation into heat [66]).

In general, the rate of dissipation of the system in its most probable stationary states will depend on the incident spectrum, the quantum efficiencies for molecular dissociation and the quantum efficiency for dissipation to the ground state through a conical intersection, $q_{j}^{IC}$. For example, if hard, dissociating, UV-C photons (λ<205 nm) were copiously available at Earth’s surface, then evolution may be towards concentration profiles of molecular dissociation products and not molecular dissipative structuring products (Figure 7), as probably occurred at some epoch on the surface of Mars. The dissociation energy of hydrogen cyanide (HCN) in gas phase is 129 kcal/mole, corresponding to a photon wavelength of 222 nm, while that of cyanogen (NCCN) is 145 kcal/mole, corresponding to a photon wavelength of 197 nm [67].

**Figure 7 entropy-25-01059-f007:**
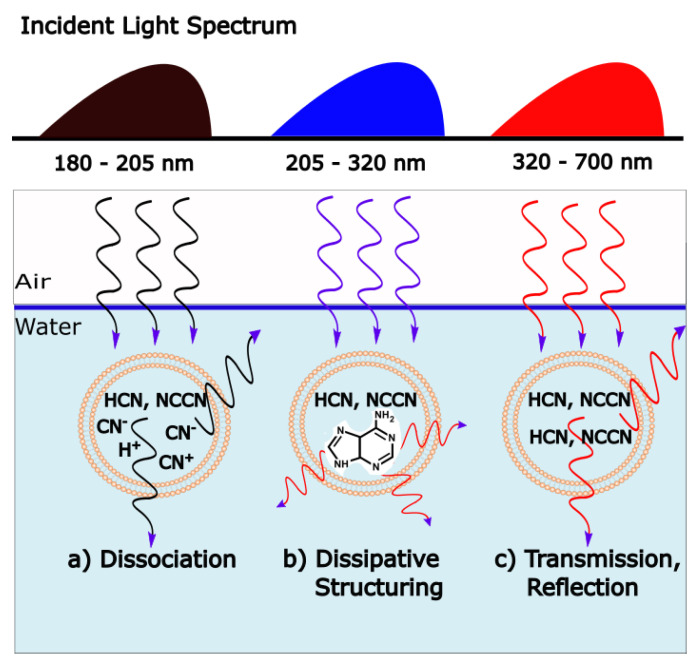
Different wavelength regimes led to different irreversible processes at the origin of life. (**a**) Hard UV-C wavelengths (<205 nm) lead to dissociation of the precursors hydrogen cyanide (HCN) and cyanogen (NCCN). Entropy production here results from the increase in structural entropy of the disassociated fragments in water and some reflection and transmission of the incident photons into a greater solid angle [66]. (**b**) Soft UV-C wavelengths (205 < λ < 280) and UV-B (280 < λ < 320), with energies of the order of carbon covalent bonding, induce dissipative structuring of the fundamental molecules of life (e.g., adenine). Entropy production here results from the dissipation the strongly absorbed incident photons into heat through the conical intersection of the dissipatively structured molecules. (**c**) UV-A and visible wavelengths do not have enough energy to reconfigure carbon covalent bonds, so the precursors remain intact. Here, entropy production results from the transmission and scattering of the incident photons into a greater solid angle [66]. Continued dissipative structuring in the soft UV-C regime (**b**) over the early evolutive history of life on Earth eventually led to more complex biosyntheic pathways (and ozone shielding) allowing carbon covalent bonds to be reconfigured with the lower energy photons of regime (**c**) by combining the energy of two or more photons (e.g., ATP production derived from visible photosynthesis). This, however, occurred some hundreds of millions of years after the origin of life occurring in regime (**b**).

Since for non-linear photochemical systems under the soft UV-C light flux, many stationary states with different pigment concentration profiles, and therefore different photon dissipative efficacy, could exist, external and internal fluctuations cause the system to roam over the available stationary states, the most visited being those states with molecular concentration profiles of molecules having the largest conical intersections for rapid internal conversion to the ground states [25].

The photochemical and chemical reactions involved in the dissipative structuring of the nucleobases adenine and guanine from the precursors HCN, NCCN, and water have been given in detail in reference [25] and [33] respectively. Evolution towards these molecular concentration profiles of lower entropy is not possible if only thermal reactions occur within an isolated system, since an isolated system is obligated by the second law to increase its entropy. The origin of life, therefore, was contingent upon the Archean soft UV-C photon potential from the external environment. Subsequent evolution of biology to even greater complexity and dissipation, also depended upon the external photon potential, but with ever increasing displacement of absorption towards the region of higher photon intensity, but less energetic, UV-A and visible wavelengths (see caption of Figure 7). This implied new mechanisms of thermodynamic selection giving rise to more complex dissipative structuring, which will be described in the following section.

## 5. Natural Thermodynamic Selection at Higher Levels

Biotic levels higher than the molecular level indicate historical evolutionary trends towards greater complexity with commensurate greater photon dissipation. Examples include; nucleic acid—amino acid association (codons and the stereochemical era [68]), increase in pigment diversity, endosymbiosis, nervous systems with sensory perception and intelligence, symbiosis and mutualism among species, ecosystem succession, a general increase in species diversity over time [69], the coupling of life to abiotic processes such as the water, carbon, and nitrogen cycles, life-induced changes to Earth’s atmosphere and surface [70], and human societal and technological evolution.

Traditional evolutionary theory tacitly assumes that the evolutionary dynamics observed at these higher levels is simply the emergent result of Darwinian natural selection acting on reproductive success operating at the organismal level. If this were the case, global trends, such as those enumerated in the Gaia hypothesis, e.g., surface temperature control (to the liquid water regime in spite of increasing solar output), atmospheric transparency to visible wavelengths, limitation of atmospheric oxygen concentration and ocean pH levels and salt concentration, would have to be considered as mere oddities.

If selection was acting only at the level of the organism, other phenomena would also need explanation. For example, why would pigments continually arise over time to cover ever more of the solar spectrum even though large regions of this spectrum are not used in photosynthesis (or in any other physiological process of utility to the organism) but instead the light is simply dissipated directly into heat [36]? Why would organisms expend resources to make and exude pigments such as mycosporine-like amino acids and isoprenes into their environment without apparent benefit to the organism [45]? Why would the organismal level be the “chosen” level for selection, rather than, for example, Dawkin’s suggestion of selection at the gene level [8]? What could be the physical-chemical explanation for the “selfishness” of the genes in the Dawkin’s proposal, or the physical-chemical explanation of a presumed innate “vitality” (will to survive and reproduce) in the neo-Darwinian perspective?

If, on the other hand, it is argued that evolution at the higher levels is not a strictly emergent phenomena, but that a kind of Darwinian natural selection is also acting at higher levels, then this would behoove us to identify a plausible fitness function for selection at these higher levels. However, even if efforts were successful in delineating a viable fitness function for differential survival or reproductive success of these higher order entities, the competing population numbers dwindle going up the biological hierarchy, until, at the highest level of the biosphere, a paradox arises which can be paraphrased as “the evolution of a system of population one” [9]. There is little doubt that the biosphere is, and has been, evolving (e.g., grown and become more complex and diversified over the history of life on Earth) however, the argument that the traditional evolutionary paradigm is the driver of evolution at this level looses all interpretation because the biosphere is not in competition with any other. The traditional evolutionary paradigm is plagued with problems and paradoxes at all biotic levels.

The non-equilibrium thermodynamic paradigm, instead, suggests that evolutionary process occurring in nature at all hierarchal levels, including molecular structuring, replication, complexation, endosymbiosis, societies, biotic-abiotic coupling, are all examples of dissipative structuring and are all contingent on increasing dissipation. The greater the rate of dissipation of the imposed solar photon potential, the greater the stability, or thus probability of the processes (see Section 4). Such irreversible processes “spontaneously” arise as dissipative structures in nature driven by the entropic forces of dissipation of an externally imposed generalized thermodynamic potential and lead to internal flows which, themselves, give rise to new internal forces. For example, the impressed photon potential gives rise to chemical potentials in plants, which then becomes relevant to herbivores, and the chemical potential of the herbivores relevant to carnivores. It will be shown in the following sections how each of these components of the biosphere couples in a catalytic way to increase global photon dissipation.

The dynamics of the evolution of the biosphere can also be described in non-equilibrium thermodynamic terms. For non-linear systems under an impressed thermodynamic force or potential, various stationary states with different rates of photon dissipation, with only local stability, are available to the system for the same initial and boundary conditions. Fluctuations near critical points (boundaries of neighboring attractors) cause the system to evolve over such states, generally towards those of greater dissipation since these states have a larger catchment basin in generalized phase space, and are more stable, as explained in the example of the dissipative structuring of adenine given in Section 4. In the following section, the mechanisms of natural thermodynamic selection are generalized to the higher biotic-abiotic levels.

### 5.1. Generalization of Natural Thermodynamic Selection

Dissipative structuring at thermodynamic stationary states in the form of molecular concentration profiles of UV-C pigments, and natural thermodynamic selection of the particularly dissipative profiles occurred “spontaneously’ at the origin of life’, driven by the thermodynamic imperative of increasing the dissipation of the soft UV-C photons arriving at Earth’s surface during the Archean. In this section the concept of natural thermodynamic selection is generalized to higher biotic levels which evolved continuously, from the molecular level at the origin of life under the soft UV-C photons to the highest level of the present biosphere under the visible photons of today.

Thermodynamic selection in the biosphere (existence for a finite time in a particular stationary state out many possible) is contingent upon stationary state stability, and this, in turn, is contingent upon photon dissipation. Those internal or external fluctuations, macroscopic or microscopic, of any of the components of the biosphere which lead the biosphere to stationary states of greater global entropy production, will most likely be amplified (e.g., auto-catalytic or positive feedback processes). In terms of thermodynamic forces and flows, a fluctuation in the biosphere may cause new thermodynamic forces to arise at any hierarchal level, giving rise to new generalized flows and the elimination of others. In this way, particular molecular concentration profiles, complexes of molecules, individual organisms, communities, species, clades, ecosystems, and biospheres arise, wax and wane, or go extinct accordingly.

Since selection is contingent on the photon dissipation of the entire biosphere, at any particular hierarchal level the biotic units do not compete with each other, or struggle against their external environment, as imagined in the traditional Darwinian perspective, but rather form part of a quasi-stable global biotic-abiotic stationary state which “competes”, on stochastic fluctuation, with other similarly available stationary states of different photon dissipative efficacy in the neighborhood of a generalized phase space (e.g., molecular concentration space at the origin of life, or species population space at the level of today’s ecosystem). Those stationary states, under the specific environmental conditions (the solar photon potential arriving at Earth’s upper atmosphere), which result in greater photon dissipation are generally more probable (stable) since they have a larger attraction basin and greater photon dissipation in this state, and thus more likely to be occupied at any given period in Earth’s history (Figure 8).

### 5.2. Selection of Complexes of Fundamental Molecules

Since the origin of life, each new dissipative level of the biosphere (dimensionality of the generalized phase space, Figure 8) was added incrementally to an existing one through, for example, a particular molecular mutation or complexation, endosymbiotic, symbiotic, parasitic, mutualistic, or abiotic coupling event, each event usually (but not always) adding to the systems overall photon dissipative efficacy. For example, at the molecular level, and at the origin of life, thermodynamic selection of the molecule was based on the rate of UV-C absorption and dissipation of the excited state energy to the ground state through a conical intersection (the quantum efficiency for internal conversion $q_{i}^{ic}$), as presented in the previous section (Figure 6).

As an example of going up the biological hierarchy, we postulate here that the addition of ribose (or deoxyribose) to the nucleobases to form the nucleosides was an example of dissipative complexation occurring “spontaneously” because this reduces the lifetimes for internal conversion to the ground state following UV-C excitation by about a factor of two for adenosine, cytidine and thymidine compared to the isolated bases [71]. The decrease in lifetime is mediated by an additional internal conversion pathway involving a proton transfer along a hydrogen bond from the sugar to base [71]. The attached sugar also changes the conical intersection most probable for internal conversion from the N9 stretch to the ring puckering [72] shown in Figure 4 which puts the maximum of absorption of the complex closer to the peak in the Archean surface solar spectrum at ∼260 nm (Figure 2). Both these effects make the nucleoside more photon dissipative than the nucleobase and ribose as separate entities [73].

Phosphates and nucleosides in aqueous solution under UV-C light gives UV-activated (phosphorylated) nucleotides [74]. Adding a phosphate group H_2_${\mathrm{PO}}_{4}^{-}$ does not provide an additional route to internal conversion for adenine monophosphate (AMP), but it does provide additional absorption at 247 nm with a small oscillator strength of 0.0064 [72] allowing the phosphate group to act as an antenna chromophore, further increasing photon dissipation efficacy of the nucleotide complex in the soft UV-C region. Additionally, nucleotides are much more soluble in water (because they become more polar) than nucleosides, and considerably more so than the nucleobases, allowing them to spread throughout the volume and thereby absorb more photons (an effect similar to hyperchromism in which absorption increases by approximately 30% in going from stacked double- to unstacked single-strand DNA [43]).

Hydrogen bonding between the complimentary nucleotides (e.g., A-T, G-C) also further reduces the excited state lifetimes [71]. This again appears to be due to a proton transfer reaction between the complimentary bases, which can give rise to DNA or RNA denaturing (see below). A more photon dissipative complex would have been more likely to increase its concentration under the Archean surface soft UV-C light, as we have seen for the concentration profiles of the molecules on route to the synthesis of adenine (Figure 6) and, as we will see, for double strand RNA or DNA (Figure 9).

The formation of phosphodiester bonds between the nucleotides (through UV-C induced phosphorylation of the nucleotides [75]) improves the stability of the nucleobases (UV-C pigments) against hydrolysis [76]), but more importantly, it provides a scaffold for the attachment of UV-C antenna molecules (e.g., codon—amino acid affinity, see Figure 9), allowing for still greater photon dissipation [44].

At the macro-molecular level, e.g., the nucleic acids DNA and RNA, we have suggested [21] an Ultraviolet and Temperature Assisted Reproduction (UVTAR) process in which the enzymeless replication of DNA or RNA could occur through UV-C induced denaturing (experimentally measured [43]) of double strand oligos in the daylight hours, and magnesium or iron ion catalyzed extension [75,77] during overnight periods of cooler sea surface temperatures. Furthermore, because of a small component (∼5%) of circularly polarized light at the ocean surface (of opposite handedness in the morning and afternoon) and the circular dichromism of DNA and RNA, UV-C induced denaturing could have given rise to, not only increased dissipation (∼30% hyperchromic effect) but also the homochirality of the nucleic acids [41] due to an increase in UV-C-induced denaturing with surface temperature [43]. Just as at the fundamental molecular level (e.g., the nucleobases), at the macro-molecular nucleic acid level, therefore, selection is also based on efficacy of photon dissipation, in this latter case through a dissipation-denaturing relation operating through the UVTAR mechanism [43] (see Figure 9). (Homchirality may also have arisen from symmetry breaking emergent behavior of the formose reaction in forming ribose [78]).

**Figure 9 entropy-25-01059-f009:**
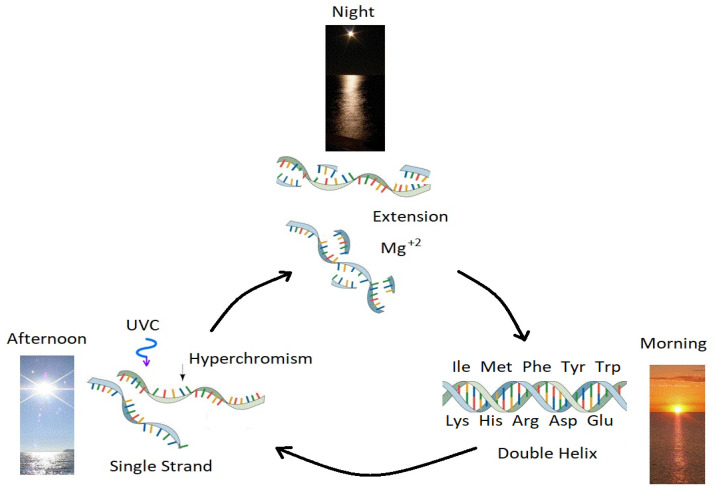
Ultraviolet and Temperature Assisted Reproduction (UVTAR) of RNA and DNA. A mechanism proposed for the enzyme-less replication of RNA and DNA assisted by the absorption and dissipation of the prevailing UV-C light flux and the high temperatures of the ocean surface during the late Hadean or early Archean, including a day/night diurnal warming and cooling cycle of the water surface due mostly to the absorption of solar infrared light. Most denaturing would occur in the late afternoon when ocean surface temperatures were highest (UV-C induced denaturing is temperature dependent [43]). Extension occurs overnight with the aid of Mg${}^{+2}$ or Fe${}^{+2}$ ions [75,77], UV-C activated nucleotides, and colder surface temperatures. “Hyperchromism” refers to an increase (∼30%) in the absorption (and dissipation) of photons at UV-C wavelengths (∼260 nm) once RNA or DNA are denatured into single strands. Oligos which had chemical affinity to the 10 amino acids listed in the figure (all of which have photon absorption and dissipation fomenting characteristics [44]), would have had a greater chance of denaturing during afternoon hours when the surface temperature was warmest, and could therefore be replicated overnight. This particular selection is only one of a number of mechanisms based on greater photon dissipation, that we have termed “natural thermodynamic selection” [21,25,32,40]. The important aspect of this auto-catalytic (template directed) mechanism is that replication is tied to photon dissipation, providing a thermodynamic imperative for reproduction and proliferation over all of Earth’s surface. Reprinted with permission from Ref. [44].

Thermodynamic selection of complexes of non-covalently bound fundamental molecules was also based on the same UVTAR dissipation-replication mechanism. For example, due to the chemical affinity of the amino acid tryptophan (an antenna molecule which absorbs strongly in the UV-C) to codons of nucleic acid [68] it could pass its photon-induced excitation energy through resonant energy transfer to the nucleic acid for dissipation through the conical intersection of one of its bases, thereby increasing overall photon dissipation. Similarly, amphipathic amino acid affinity to nucleic acid would make the nucleic acid - amino acid complex less susceptible to sedimentation and therefore more exposed to the surface UV-C light. Since greater absorbing nucleic acid complexes are more likely to experience UV-C induced denaturing [43] and therefore replicate (Figure 9), the UVTAR process would thus lead to information concerning the affinity to these antenna or amphiphilic molecules becoming selected (programmed) into the nucleic acid oligos [44]. This would lead to a stereochemical era for amino acid - nucleic acid association, for which there appears to be ample evidence [68]. This could also be considered as an early example of higher level thermodynamic selection, or “group selection” (codon plus amino acids), again based on increasing the efficacy of photon dissipation.

We can also take this complexation through thermodynamic selection another step further to the stereochemical association of the dissipative nucleic acid - amino acid complex with dissipatively structured fatty acid vesicles [42]. Besides diurnal variations of the sea surface environment due to Earth’s rotation, there would also be seasonal variations due to axial tilt of Earth and its elliptical orbit around the sun. The ocean surface was also cooling throughout the Archean epoch. Thus, depending upon the latitude, or the epoch, some regions of the ocean surface may have become too cold in winter to allow the UVTAR mechanism to be effective. In these regions, DNA or RNA oligos with affinity to fatty acid vesicles may have become favored since the vesicles provided an insulating enclosure where UV-C light could enter and be absorbed on the fundamental molecules, but the heat of dissipation would have remained trapped for some time (due to the low heat conductivity of the hydrocarbon tails of the fatty acids), raising the temperature within the vesicle enclosure sufficient to allow the UVTAR denaturing process to occur with greater probability on colder seas.

The association of the nucleic acid - amino acid complex with fatty acid vesicles may have been through the positively charged amino acids lysine, arginine, and histidine, which all have strong stereochemical affinity to their codons/anticodons [44] and a positive charge to attach to the deprotnated, negatively charged, head group of fatty acids (fatty acid vesicles form at a pH near to the pKa of their headgroup because interactions between the protonated and deprotonated forms of the headgroup stabilize the structure). Neutralization of the negative charges on the RNA or DNA backbone by either salt ions or positively charged amino acids would also have allowed overnight extension of nucleic acid to occur at higher surface temperatures than otherwise [44].

Fatty acid vesicles thus provided, (i) an enclosure for concentration of the components, through differential permeability, facilitating further chemical and photochemical interactions [25], (ii) continued UVTAR denaturing and replication in colder seas through the trapping of the heat of photon dissipation within the vesicle (Figure 9), and (iii) the shielding from hard (short wavelength) UV-C photons that could ionize or disassociate the molecular complexes [79]. The stereochemical information for this nucleic acid + amino acid + fatty acid vesicle association would thus also gradually become programmed into the DNA through the UVTAR mechanism based on photon dissipation, in the same way as it was for the nucleic acid—amino acid association (Figure 9) [44].

In order that the concentrations of the fundamental molecules within the vesicle increase sufficiently to allow their polymerization into nucleic acids, the protocell vesicle walls would have to be permeable to the small precursor molecules, but impermeable to large dissipatively structured molecules like nucleotides and RNA or DNA oligos. The lipid membranes would also need to have the characteristic of growth and binary fission triggered, eventually, on RNA or DNA replication. The the greater stability of the nucleic acids (compared to isolated nucleobases) and the facilitation of nucleic acid polymerization at higher nucleobase concentrations, plus replication through the UVTAR mechanism, all leading to increases in photon dissipation, were the thermodynamic reasons for the origin and replication of the vesicles containing nucleic acid, i.e., the first protocells [25,42].

### 5.3. Evolution of Chemotrophic Organisms as Catalysts for Phototrophic Organisms

The solar photon potential drove most of the dissipative structuring known as life. However, today there also exist chemotrophic organisms (e.g., bacteria living at hydrothermal vents and underground, and animals) that dissipate a chemical potential. Most of these chemical potentials are secondary, resulting from processes (e.g., photosynthesis) that dissipate the solar photon potential. For example, most bacteria underground or at the bottom of the oceans are dissipating the chemical potential available from buried organic material or the rain of organic material from the surface, respectively. Animals dissipate the chemical potential that was originally produced in plants and cyanobacteria.

There also exist, however, chemolithotrophic organisms (bacteria and archaea) at the surface, under water and underground, which survive independently of sunlight. They have special enzymes for oxidation of inorganic electron donors (e.g., hydrogenase, carbon monoxide oxidase, nitrite oxidoreductase, sulfite oxidase, etc). As an example, chemolithotrophic microbes, found kilometers below the ocean and land surfaces, appear to depend exclusively on geologically produced hydrogen and sulfur compounds. Hydrogen comes from the decomposition of methane by high temperatures and pressures, or the decomposition of water by the radioactive decay of uranium, thorium and potassium. All of these microbes have been shown to be genetically related to cousins on the surface, having been buried and separated from them for several million to tens of millions of years. The metabolism of these ocean or land buried organisms is 10,000 to one million times slower than at the surface. The evolution of these organisms is, therefore, also correspondingly much slower than at the surface, affected only by point mutation and genetic drift [80].

There is no evidence that chemotrophic organisms preceded phototrophic organisms. In fact, a large set of optimized amino acid sequences (enzymes) would have been needed for a primordial chemoautotrophic protocell because the chemical free energy available in inorganic substances (redox potential) is orders of magnitude smaller than the free energy available in a single UV-C photon, thus inhibiting reactions involving carbon covalent bonding which commonly have high activation barriers, unless such enzymes were already available at the origin of life. For this reason, the first “living” cell most likely arose as a dissipative structure under the soft UV-C light potential rather than under a chemical potential. Evidence for this was presented in the extraordinary UV-C photochemical properties found for the fundamental molecules of life listed in Section 3.1.

The chemotrophic organisms, although unlikely primordial, are not irrelevant to photon dissipation. Through their metabolism, excrement, and death, they provide a nutrient recycling service to the phototrophic organisms. In our thermodynamic view, they are thus, first and foremost, catalysts for the phototrophic organisms dissipating photons.

The enzymes of these chemotrophic organisms were selected over millions of years of evolution almost exclusively according to how they improve their catalytic effectiveness in aiding photon dissipation. The mechanism of this thermodynamic selection of enzymes leading to increased global photon dissipation, is symbiosis (cross catalysis). As an example, the greater the ability of underground rhizobacteria bacteria to provide the required nutrients to plants (e.g., nitrogen) the greater the number of chlorophyll and other pigment molecules and thus the greater the photon dissipation rate in the leaves of the plants and therefore the greater the osmotic pressure to draw up water with nutrients from the roots, allowing the symbiotic bacteria to acquire chemical potential and photosynthetic carbohydrates from plants for their own maintenance. Gene selection for both the plant and the chemotrophic bacteria is thus optimized for global photon dissipation, not for the differential reproductive success of either one of the symbionts. Growth, the increase in numbers of phototrophic organisms and their spread over the whole of Earth’s surface (traditional evolutionary perspective), is just one of many ways that global photon dissipation is increased. Others include, (1) increases in the region of wavelength absorption of plant pigments to cover the whole solar spectrum, (2) increasing the rate of photon-induced excited state dissipation to the ground state, allowing the pigment to process (dissipate) more photons, (3) coupling of the dissipation of photons in the leaves to the water cycle and other abiotic processes (shifting the wavelength of the outgoing photons even further towards the infrared), (4) exudation of pigments into the environment (e.g., oxygen, mycosporine-like amino acids, isoprenes, etc.). All of these factors important to photon dissipation are “optimized” through the natural thermodynamic selection of enzymes of both photoautotrophic and chemotrophic organisms.

### 5.4. Symbiosis as the Mechanism of Selection for Photon Dissipation at the Ecosystem Level

The symbiosis of different cellular organelles, and the parasitic, symbiotic, or mutualistic interaction among species from all three domains of life, and the coupling of biotic with abiotic dissipative processes, can all be associated with increasing biosphere photon dissipation through increases in efficacy of plant and cyanobacteria photon dissipation and the spread of organic pigments over the whole of Earth’s surface [36,70,81]. An example of this thermodynamic selection through symbiosis at the organismal level (plants and rhizobacteria bacteria) was given in the previous subsection. An example at the ecosystem level is the re-introduction of wolves into their historical homelands in Yellowstone National Park after uncontrolled hunting led to their extinction in 1926. On re-introduction in 1995, these top predators kept the deer and elk populations always on the move, thereby preventing them from overgrazing and helping to spread their excrement and dead carcases as nutrient over a wide area. This led to a general greening of the park [82], which means greater photon dissipation. This is an example of how natural thermodynamic selection operates at the ecosystem level, bringing it to climax states (stationary thermodynamic states) of ever greater photon dissipation (Figure 8). Similarly, fish and whales in the oceans bring nutrients from the depths and deposit them as excrement and dead carcases at the ocean surface, even far from shore where little nutrients are normally found, fomenting in this way replication of the photon dissipating algae and cyanobacteria.

### 5.5. The Biosphere and Its Human Perturbation

This coupling of irreversible processes is repeated at each biological level, incorporating, as well, abiotic dissipative processes, creating the greatest dissipative process of all, the biosphere of today. For example, the water cycle is coupled through the heat of dissipation of photons in organic pigments in the leaves of plants or within cyanobacteria on the surfaces of the oceans, lakes, and wet soils. The coupling of the water cycle to photon dissipation is also autocatalytic since more water in the water cycle implies a greater greening of Earth [70,83]. The water cycle dissipates the infrared light of the heat of photon dissipation in the leaves even further towards the far infrared, finally emitting the energy into space at the cloud tops having an approximate black-body temperature of −14 °C (corresponding to an emission peak at ∼11 μm).

An example of thermodynamic selection involving the human species, operational at the societal level, is the human-induced increase in atmospheric CO_2_ since the industrial revolution. This, surprisingly, has also led to an important greening of Earth, for example, increased growth rates of algae, cyanobacteria, and vegetation, and high albedo glaciers replaced by low albedo forests [84]. Although too much atmospheric CO_2_ may lead to ecosystem collapse through, for example, a run away greenhouse effect, natural thermodynamic selection will eventually assert itself in no uncertain terms, as indeed seems to be happening, for example through natural catastrophes (e.g., drought associated with global warming) by limiting the activity of humans and thus Earth can be expected to arrive at a new stationary state with an atmospheric concentration of CO_2_ concordant with greater photon dissipation and human activity. This sounds much like the Gaia hypothesis [11], except that the fitness function being optimized is not the “suitability of Earth for all life”, but rather global photon dissipation.

Finally, since today’s biosphere has both biotic and abiotic components coupled on many different levels and over different time scales, it is relevant to make a few remarks concerning the coupling of biotic and abiotic irreversible process. Both biotic organisms and abiotic processes have the ability to adapt to a changing impressed thermodynamic potential, implying that thermodynamic selection is a finite time process. Biotic organisms today adapt through their organismal plasticity (e.g., the ability to migrate, the ability to survive off different thermodynamic potentials, i.e., heterotrophy, or through mutation of their genes and reproduction, i.e., plasticity at the species level). In contrast, abiotic processes have an inherent plasticity, for example, a change in size or direction of a hurricane in response to a change in the size or direction of the ocean surface temperature, steered, in fact, by cyanobacteria concentration [85], demonstrating yet another biotic-abiotic coupling.

Neo-Darwinian evolutionary theory suggests that life struggles to survive against an imposing external environment while competing with other organisms. Gaia theory speaks of “life shaping the environment for its own suitability”. Our non-equilibrium thermodynamic perspective suggests that irreversible processes are not “struggling to survive”, rather, they are instead thermodynamic flows which rise and fall in response to changes in internal thermodynamic forces as fluctuations take the global system to different stationary states, in the general direction of greater stability, which under the impressed photon flux, corresponds to increasing photon dissipation (Figure 8).

The thermodynamic dissipation theory speaks of life coupling at many different levels to other irreversible thermodynamic processes (both biotic and abiotic) which are together evolving to ever greater levels of biosphere photon dissipation (entropy production) under the prevailing solar photon potential. It is this impressed photon potential, together with non-equilibrium thermodynamic principles (in particular the second law and the conservation laws), that are the creators, transformers, and selectors of those irreversible dissipating processes at all biotic levels we associate with life.

The biosphere or ecosystem can thus be described by a matrix of coefficients representing the interaction between different biotic and abiotic dissipative processes, associated, for the most part, with photon dissipation. These interaction coefficients evolve not to optimize the survival of the representative irreversible process (the gene, individual, species, ecosystem, or the biosphere) but instead to optimize the photon dissipation of the entire biosphere. Being the biosphere a non-linear dissipative system, the allowed population dynamics at the stationary states can be either point, cyclical, or chaotic attractors [20].

The information required for the building of, and maintaining quasi-stability of, the entire biosphere, including its abiotic components (such as the water cycle, the transparency of the atmosphere, oxygen levels, sea water, salt and pH levels, ocean currents, winds, surface temperature, etc.), becomes, in this way, encoded into the collective genomes of all biotic organisms. Complex biotic dissipative structures, such as animals, trees, or societies, having evolved through this process of genome information accumulation related to photon dissipation, store information about the generalized chemical potentials existing in their present and past environments. This information endows them with a plasticity to “adapt” to different chemical potentials by building on existing structures (e.g., the Archean UV-C dissipative nucleobase pigments used today as letters in codons to store information in the nucleic acid), or to rapidly return to dissipation of potentials that have come back to the environment after a certain absence, thus allowing them to evolve into ever more numerous, efficient, and adaptive dissipative structures. Natural thermodynamic selection still acts at the molecular level today by selecting pigments in the leaves of the plants and cyanobacteria that are most efficient at absorbing and dissipating the photon flux into heat, and new pigments will arise in the future, pushing photon dissipation ever further from the “red edge” of today towards the infrared.

## 6. Resolution of Problems and Paradoxes Inherent in Traditional Evolutionary Theory

The non-equilibrium thermodynamic paradigm provides resolution of some existential problems and paradoxes inherent in traditional evolutionary theory. First and foremost, the thermodynamic paradigm provides a foundation for evolutionary theory based on well established physical law. The conservation laws and the second law of thermodynamics incorporated into classical irreversible thermodynamic theory in the non-linear regime are sufficient to describe the general dynamics of dissipative structuring and evolution of a material system interacting with its environment. Such a system evolves over the entropy production surface of a multi-dimensional generalized phase-space through perturbation near critical points (points on the border between neighboring stationary state attractors), leading generally, particularly for positive feedback (autocatalytic) systems, to greater dissipation of the impressed environmental potential (e.g., the solar photon flux at the level of the biosphere).

Unlike traditional evolutionary theory, in which the origin of life remains an enigma, the dissipation paradigm provides a unified framework from within which both the origin of life and its subsequent evolution can be explained. First, the initial dissipative structuring of UV-C pigments to dissipate this region of the solar spectrum prevalent at Earth’s surface throughout the Archean. Dissipative structuring with thermodynamic selection led to greater molecular complexation which increased photon dissipation. Finally, after the invention of oxygenic photosynthesis leading to the exudation of the UV-C pigments oxygen and ozone into the atmosphere, shielding the surface from UV-C, the dissipative structuring of the visible pigments through more delicate (non-covalent) and complex biosynthetic pathways became possible to dissipate the higher intensity visible wavelengths.

There is no “survival of the survivors” tautology in thermodynamic theory since there is an explicit “fitness function”, biosphere photon dissipation, that increases stochastically over time because such dissipative stationary states are generally more stable to internal and external fluctuation. There is also no need to assume a“vitality” inherent in biotic material. All material will dissipatively structure, depending on the strength of the atomic bonding and appropriate wavelength region. Life started as the dissipative structuring of pigments from carbon covalent bonded material under soft UV-C light.

There is no single level on which natural thermodynamic selection operates. It acts simultaneously on all levels, selecting, generally, but not always (through statistical fluctuation), stationary states of the biosphere of generally greater photon dissipation. At the highest level of the biosphere, the paradox of “the evolution of a system of one” is resolved since greater biosphere photon dissipation is a valid fitness function for thermodynamic selection, allowing evolution at this level under perturbation.

The paradox of the first tier is resolved since thermodynamic selection is not competition between species leading towards “better fit” species, but rather increases in photon dissipation and this is often obtained by building on previous dissipative structures and re-purposing, as we have seen for the fundamental molecules of life which began as UV-C pigments but are today information carrying molecules. The archea and bacteria of the first tier still exist today and are also found as the organelles and symbionts in higher level organisms like the eukaryotes. They were not “out-competed”, their pigments still form the foundations of photon dissipation and they have also been incorporated into systems at new levels of dissipation.

Punctuated equilibrium, a hallmark of biological evolution [7], without explanation in the traditional evolutionary paradigm, can now be understood as the statistical evolution through different stationary states on the entropy production surface of the non-linear non-equilibrium system, promoted by fluctuations close to critical points (borders of stationary state attraction basins). Selection, related to dissipative stability, is thermodynamic and involves both the system and its environment through the coupling of biotic with abiotic irreversible processes.

At the level of the organism, our thermodynamic theory resembles Darwinian theory since the greater the number of dissipative units (organisms), usually (but not always) the greater the global photon dissipation. Each organism is a dissipative process contributing either directly or catalytically to the global entropy production. Reproductive success at the level of the organism is thus often well correlated with photon dissipation. However, dissipative efficacy entails not only numbers, but also other photochemical characteristics, and the coupling of species in non-linear ways to other species and to the abiotic irreversible processes (such as the coupling of the heat of photon dissipation produced in organic pigments to the water cycle). All of these contribute to photon dissipation and entropy production. Selection is therefore not truly based on reproductive success of the individual, but rather on the global photon dissipation rate of the whole system, which corresponds to the system being more often found at the larger peaks with wider attraction basins on the entropy production surface of the generalized phase space (Figure 8).

At the level of the biosphere, our thermodynamic theory resembles Gaia theory in that there is a coupling of biotic processes with abiotic processes. Thermodynamic stationary states have local stability and small fluctuations will tend to be dampened leading to a local homeostasis, as emphasized in Gaia theory. However, unlike in Gaia theory, selection of the abiotic characteristics is not based on “suitability for all biotic life”, but rather, again, stochastically on the global photon dissipation rate of both biotic and abiotic processes and their coupling.

## 7. Conclusions

Traditional evolutionary theory can only be regarded as a heuristic narrative, useful for making some sense of the biological and paleontological data concerning evolution at the level of the organism. The theory, however, lacks a chemical-physical foundation which leaves it with many problems and paradoxes. Even its most conscientious proponents consider the theory as, at the very least, “incomplete”. Here I have presented a different perspective based on non-equilibrium thermodynamic theory providing an understanding of material interaction with its environment based on the conservation laws and the second law of thermodynamics. In this paradigm, material under an impressed thermodynamic potential organizes “spontaneously” in such a manner so as to dissipate more efficiently the imposed potential. The mechanisms of organization and selection leading to evolution are different at different biotic-abiotic levels, but all are based on stochastically increasing dissipation, and the process of evolution towards more complex systems is known as “dissipative structuring”. Dissipation is the only agent of organization.

The Archean soft UV-C photon potential is identified as the thermodynamic potential (impressed force) giving rise to the resulting flow (photochemical reactions) corresponding to the dissipative structuring of the fundamental molecules (UV-C pigments) at the origin of life. The propensity of UV-C light to form conjugations in carbon based molecules, which promotes photon absorption and dissipation, is probably the most important reason why life is based on carbon, and not other similar outer electron shell elements like silicon. A specific example was given of the dissipative structuring of adenine. Each additional molecular complexation which occurred (e.g., nucleic acid - amino acid association) would have been selected on greater photon dissipation because this implies greater probability (stability) under the photon flux.

Many stationary states, consisting of different molecular concentration profiles, are available to such a non-linear system, and the number of these states grows as the size of the system grows, each new layer increasing the photon dissipation capacity of the growing biosphere. Evolution to greater photon dissipative stationary states is stochastic, dependent on fluctuations near a non-equilibrium thermodynamic critical point (borders of the stationary state attraction basins), but guided by thermodynamic law (the conservation laws and the second law of thermodynamics). Stationary states with greater photon dissipation (e.g., catalytic and autocatalytic) are generally more stable since these have larger attraction basins in a generalized phase space and are those most efficient at dissipating the free energy in the impressed potential which is available for change.

In today’s biosphere, thermodynamic selection is still based on photon dissipation, as it was at the origin of life. Today, however, with no UV-C light arriving at Earth’s surface, the possibility of direct, single photon, permutation of carbon covalent bonds to produce new dissipatively structured pigments no longer exists, and this is probably why life de novo has not appeared since the Archean. This also implies that there would be no possibility for carbon based life arising on planets of other stars lacking the important soft UV-C component in its spectrum (such as most M-type red dwarf stars).

The biosynthetic pathways of today have evolved under the principle of increasing photon dissipation to be necessarily more complex in order to also dissipate the lower free energy, but higher intensity, available in visible photons. Evolution of the biosynthetic pathways to produce the pigments oxygen and ozone were necessary before this stage of life could occur since these blocked the UV-C light from the more delicate biosynthetic pathways now incorporating not only covalent bonding, but weaker van der Waals, ionic, and hydrogen bonding.

Life is not exclusively ordered through dissipation of the solar photon potential, but this potential is by far the most important, all others being either obtained secondarily from the photon potential or very rare (e.g., chemolithotrophy). The evolutionary process of the complexation of life would thus most likely have gone through the following major stages of dissipative structuring; (1) UV-C pigments (fundamental molecules), (2) photon dissipative molecular complexes, (3) protocells consisting of UV-C dissipative molecular complexes within vesicles, (4) exudation into the environment of the pigments oxygen, mycosporine-like amino acids and isoprenes, to dissipate UV light, shielding the more complex but delicate enzymes (e.g., van der Waals and hydrogen bonded folded proteins), allowing the dissipative structuring using, and the dissipation of, visible photons (photosynthesis), the Krebs cycle, as well as chemolithotrophy, (5) symbiosis, parasitism and mutualism, (6) coupling of biotic and abiotic dissipative processes, (7) intelligence and technology to spread plant nutrients (e.g., phosphorus and CO_2_), to dissipate underground stored chemical potential (e.g., coal and petroleum), and to bootstrap carbon-based photon dissipation on other planets.

Thermodynamic selection today, therefore, is still based overwhelmingly on increasing photon dissipation, with a much smaller part attributable to bacterial and animal dissipation of chemical potentials. Bacteria and animals, however, play a much greater thermodynamic role in producing and spreading nutrients over the whole surface of Earth, thus fomenting the growth and propagation of the photon dissipating plants and cyanobacteria. Humans have unwittingly played an important role by greening the planet through the dissipative structuring of a number of societal and cultural processes leading to a greater spread of phosphorus on land and increasing CO_2_ in the atmosphere, and it appears not to be long before we begin to do the same on other planets of our solar system.

## Figures and Tables

**Figure 1 entropy-25-01059-f001:**
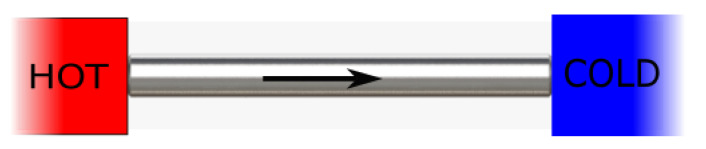
An example of a thermodynamic stationary state corresponding to a constant flow of heat from an infinite hot reservoir to an infinite colder one along a metal bar. Before the bar is connected to the heat reservoirs, local temperatures along the entire bar are the same, and we say the metal bar is in *thermodynamic equilibrium* with its environment. Connecting the bar to the heat reservoirs takes the system out-of-equilibrium, and the temperatures at each point along the bar begin to change, but eventually reach fixed values at which point the flow of heat and the entropy production come to constant values. The non-equilibrium system has now arrived at a stable “stationary state”. The generalized thermodynamic “force” is the spatial gradient of the temperature across the bar divided by the local temperature, and this is often linearly proportional to the corresponding “flow” of heat (Fourier’s law). The linearity between the flow and the force implies that, in this case, the stationary state is unique and globally stable. Each macroscopic, but small, region along the bar will be in local equilibrium and so will have a local temperature. Classical Irreversible Thermodynamic theory is therefore applicable and we can use the same variables, and the same Gibb´s relation among them, as we use in equilibrium thermodynamics, but now the variables become both space (position along the bar) and time dependent.

**Figure 2 entropy-25-01059-f002:**
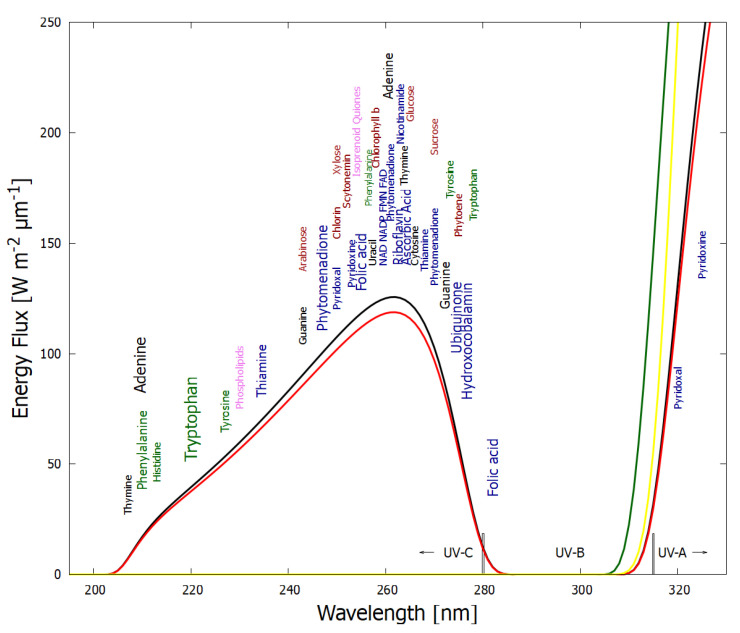
The spectrum of UV light available at Earth’s surface before the origin of life at approximately 3.9 Ga and until at least 2.9 Ga (curves black and red respectively), perhaps even extending throughout the entire Archean until 2.5 Ga [34]. Atmospheric CO_2_ and probably some H_2_S were responsible for absorption at wavelengths shorter than ∼205 nm, and atmospheric gas aldehydes (e.g., formaldehyde and acetaldehyde, common photochemical products of CO_2_ and water) absorbed between about 280 and 310 nm [35]), approximately corresponding to the UV-B region. By around 2.2 Ga (green curve), UV-C light at Earth’s surface was completely extinguished by the UV-C pigments oxygen and ozone resulting from organisms performing oxygenic photosynthesis. The yellow curve corresponds to the present surface spectrum. Energy fluxes are for the sun at the zenith. The fundamental molecules of life, suggested to have been dissipatively structured under this light as UV-C pigments, are plotted at their wavelengths of maximum absorption; nucleic acids (black), amino acids (green), fatty acids (violet), sugars (brown), vitamins, co-enzymes and cofactors (blue), and pigments (red) (the font size is roughly proportional to the relative size of the respective molar extinction coefficient). Adapted with permission from Ref. [36].

**Figure 3 entropy-25-01059-f003:**
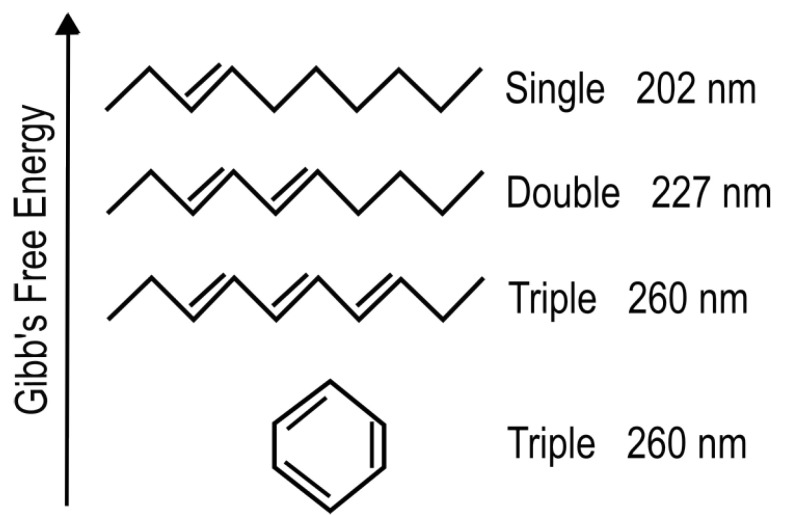
Conjugated carbon molecules are more stable (lower Gibb’s free energy in the ground state) but more importantly provide collective molecular orbitals giving rise to excited states at energies adequate for the absorption of soft UV-C photons arriving at the Archean surface. The greater the conjugation, the greater the wavelength of maximum absorption. The wavelength of maximum absorption can, therefore, easily be tuned by a simple protonation or deprotonation event. Conjugation is also important for giving the molecule a conical intersection allowing rapid dissipation of the excited state energy into heat (internal conversion).

**Figure 6 entropy-25-01059-f006:**
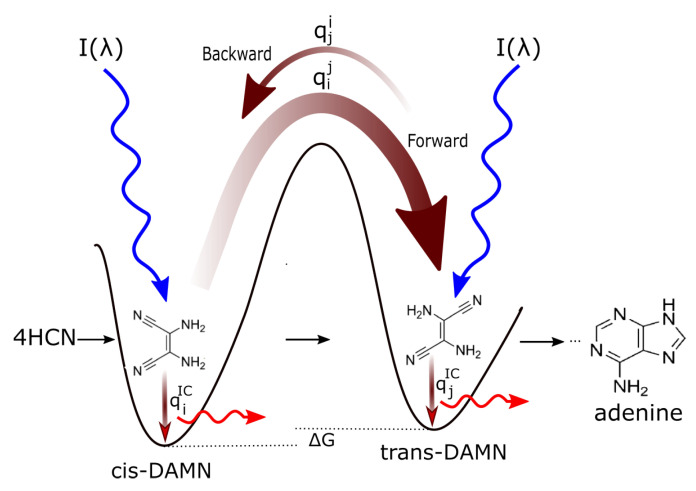
Mechanism for the evolution of molecular structures towards ever greater photon dissipative efficacy (microscopic dissipative structuring) on route to the fundamental molecules (in this case adenine, see Figure 5). The high activation barriers between configurations mean that reactions will not proceed spontaneously but only through coupling to photon absorption events. Forward and backward rates depend on photon intensities I(λ) at the different wavelengths of maximum absorption for the two structures, and on the phase-space widths of paths on their excited potential energy surface leading to the conical intersection giving rise to the particular transformation, implying different quantum efficiencies for the forward (q${}_{i}^{j}$) and backward (q${}_{j}^{i}$) reactions. Assuming that the intensity of the incident spectrum is constant, and since $q_{i}^{j}+\cdots q_{i}^{IC}=1$ and $q_{j}^{i}+\cdots q_{j}^{IC}=1$ (where the “⋯” represents quantum efficiencies for other possible molecular transformations), those stationary states (corresponding macroscopically to molecular concentration profiles) with greater photon dissipative efficacy (higher quantum efficiency for internal conversion q${}_{j}^{IC}$) will therefore gradually become more predominant under a continuously impressed UV-C photon flux, independently of the sign or size of the difference in the Gibb’s free energies ΔG of the molecules. This process, of selection of molecular concentration profiles of ever greater photon dissipative efficacy, driving evolution towards the right in the diagram, we call *natural thermodynamic selection*.

**Figure 8 entropy-25-01059-f008:**
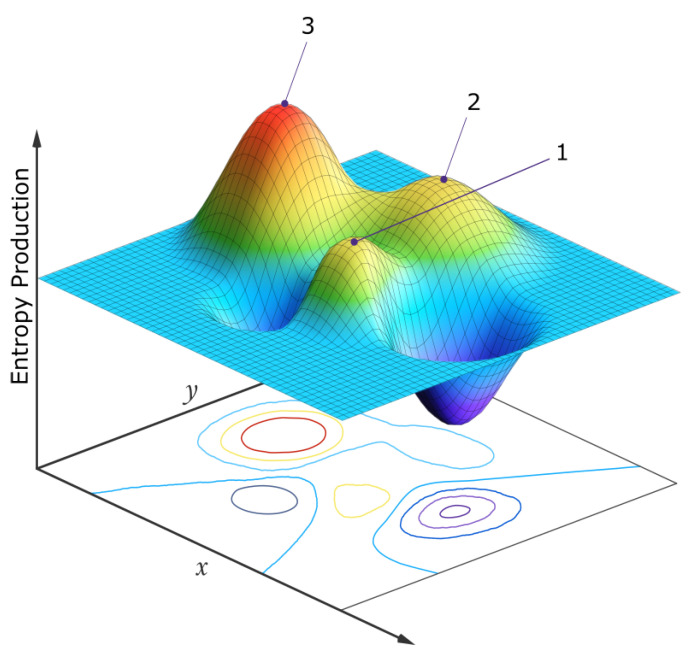
A simplified 2-dimensional schematic representation of the entropy production surface (EPS) of a generalized phase space for a biosystem under a constant solar photon potential. The variables *x* and *y* at the origin of life may be, for example, the concentrations of different pigment molecules, while for an ecosystem of today, the variables may be the populations of different species. Three locally stable stationary states at local peaks in the entropy production surface are presented. On large enough external or internal perturbation, the system evolves from one stationary state to another. Although fluctuations are generally stochastic, the system will most often be found in those stationary states with a larger attraction basin and with a higher peak in photon dissipation (the stationary state labeled “3”). For molecules, this corresponds to concentration profiles with greater quantum efficiency for dissipation to the ground state through a conical intersection. For an ecosystem, this corresponds to animal and plant population profiles giving greater total photon dissipation (time local *climax* ecosystems). If the system began in stationary state 1, its most probable future evolution would be 1 → 2 → 3, but any combination is possible. For the biosphere, the *x* and *y* variables might be the number of species in two different clades and sub-peaks (not shown) corresponding to different species populations would exist on the main peaks and evolution would usually be local, among the sub-peaks, but every once in a while a perturbation may be large enough (for example, an asteroid impact) to move the system from one main peak to another (e.g., 1 → 3, mammals *y* becoming more prominent than dinosaurs *x*). Point, cyclic, or even chaotic dynamics are allowed superimposed on these peaks [20]. Autocatalytic stationary states have higher peaks and larger attraction basins in this generalized phase space and are thus more probable. The dimensionality of the generalized phase space is not fixed but evolves over time providing new “shorter” routes to larger peaks of entropy production (e.g., the re-introduction of a population of wolves into the ecosystem of Yellow Stone National Park, leading to a greening of the park, see text below). Adapted from Ümit Kaya via LibreTexts (CC BY-NC).

## Data Availability

No new data were created or analyzed in this study. Data sharing is not applicable to this article.

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
