# Peer review of "The Non-Equilibrium Thermodynamics of Natural Selection: From Molecules to the Biosphere"

_entropy, 2023, doi:10.3390/e25071059_

Round 1

Reviewer 1 Report

The fundamental aim of the paper is to address a physical basis to the natural selection.

This aim is very interesting.

I can suggest some improvements:

- Eq. (6) must be discussed, with particular regards to its physical fundametal, because the entropy is introduced in relation to a quantum disequality (Heisenberg), but entropy isn't defined for an atom or a molecules, as well as temperature. It needs more detailed;

- An entropic approach has been developed since 2015 by Lucia et al. and I think that it must be considered in this paper;

- A definition of the thermodynamic system must be better introduced: which is the boundary? The system is open, closed, isolated? How to write the balance queations?

After having introduced these improvements, I think that the paper could be accepted.

Author Response

I thank the reviewer for their kind remarks and for comments that have helped improve the manuscript. The reviewer's comments and queries are in bold face in the following. 

The fundamental aim of the paper is to address a physical basis to the natural selection. This aim is very interesting. I can suggest some improvements: - Eq. (6) must be discussed, with particular regards to its physical fundametal, because the entropy is introduced in relation to a quantum disequality (Heisenberg), but entropy isn't defined for an atom or a molecules, as well as temperature. It needs more detailed; - An entropic approach has been developed since 2015 by Lucia et al. and I think that it must be considered in this paper; - A definition of the thermodynamic system must be better introduced: which is the boundary? The system is open, closed, isolated? How to write the balance queations? After having introduced these improvements, I think that the paper could be accepted. 

The section referring to the relation between the entropy production and the Heisenberg uncertainty relation has been re-written (lines 397 - 425), taking care to describe the physical system in detail (lines 402-403). Our system is a macroscopic thermodynamically open system consisting of UV-C light interacting with organic molecules in a water solvent within a fatty acid vesicle floating at the ocean microlayer. The temperature is that of the solvent. The flows into and out of the volume (defined by the vesicle) are flows of photons and precursor organic molecules. The balance equations are complex but have been explicitly written down for the dissipative structuring of adenine in a previous published paper of ours in this same edition [1]. I now make reference to this paper in the revised version (line 397). The Heisenberg uncertainty principle, of course, applies to any size system.  

A reference to Lucia 2015 has been included in the Introduction beginning at lines 95-98, together with other relevant papers dealing with the application of irreversible thermodynamics to biology. 

[1] Michaelian, K. The Dissipative Photochemical Origin of Life: UVC Abiogenesis of Adenine. Entropy 2021, 23, 217. https://doi.org/10.3390/e23020217 

Author Response

I thank the reviewer for their comments that have helped improve the manuscript. The reviewer's comments and queries are in bold face in the following. 

If one could accepts the starting hypothesis I could recommend the article for publication in Entropy. However, I have some serious problems with accepting this hypothesis. Let us accept that UV light synthesis amino acids spontaneously. So what? The amino acids are soluble and in water and the concentrations in the oceans must be so low that the time with syntheses to polypeptides will exceed the age of the Universe. But even if it succeeded, maybe in Darwin’s warm little pond, the amino acid and the peptides are if not in a racemic proportion, then they will be it later due to an active isomerization kinetics. So my problem is that even if one synthesizes biomolecules, they are unstable and in the case of optic active amino acids and carbohydrates they spontaneously racemize. The early sign of life is found in hot springs and the biosphere includes the Earth’s crust where most biomaterials are present. There is no UV-light so I am very skeptical of the basic hypothesis. In conclusion, I think that the author must give us an acceptable procedure for that a bio-synthesis on the water surface can lead to life. 

A robust explanation of the origin of life requires a clear understanding of not only how biologically important molecules spontaneously emerged, but also how they proliferated and evolved together into ever more complex dissipative structures, eventually leading to the global dissipative processes known as the biosphere, incorporating both biotic and abiotic components. What is important, therefore, is not that “UV light synthesis” is viable for many of the fundamental molecules of life, but rather that these molecules arise and become more complex through dissipative structuring under UV-C light to dissipate this light into heat, and therefore their origin and complexation into greater dissipative structures has a physical basis and a thermodynamic “reason” for being. Such an assertion of chemical-physical relevance cannot be made for the traditional evolutionary paradigm or for an equilibrium thermodynamic paradigm. 

Obtaining concentrations of the fundamental molecules sufficient to allow further processing (such as polymerization and complexation), is a problem for all theories concerning the origin of life, including hydrothermal vent theories. Our ocean surface scenario has the advantage of surface tension and the anchoring of amphipathic organic molecules to the surface due to their hydrophobic parts. This leads to organic molecule concentrations at the surface microlayer of up to 104 to 105 times larger than what they are in the bulk water only a centimeter below [1,2]. Furthermore, the concentration of organic molecules “trapped” at the surface would have continued to grow, while that anywhere else in the bulk water would eventually be destroyed after passing through hydrothermal vents every 10 million years or so. Even more importantly, in considering the UV-C dissipative structuring of adenine [3], we have shown that concentrations of this molecule could have increased 6 orders of magnitude (from initial values of 10-10 to 10-4 M) within only 60 Archean days assuming photochemical dissipative structuring within a fatty acid vesicle allowing small precursor molecules like HCN and UV-C light to enter but trapping larger product molecules within [3]. 

This same ocean surface origin scenario suggests how homochirality could have arisen from the phenomena of photon-induced denaturing of RNA and DNA [4], a phenomenon which we have experimentally measured and found to be temperature dependent [5], thus favoring for denaturing (and, therefore, subsequent replication) the particular oligos with the chirality of the UV light available in the afternoon when sea surface temperatures are highest. The size of the light-induced denaturing effect depends on the size of the optical dichroism at the wavelengths of maximum intensity in the UV-C region (260 nm), and this becomes greater if RNA or DNA are paired with opposite chirality amino acids [4]. There is also a stereochemical affinity effect of opposite handed nucleic acids with amino acids. 

If optically active amino acids are coupled with the nucleic acids, they pass their photon excitation energy to the nucleic acid (through resonant energy transfer) which has a conical intersection allowing for rapid (sub picosecond) internal conversion to the ground state [6] and, therefore, a very small probability for racemization, or any other chemical reaction.   

There is still no generally accepted solution to the chirality problem of early life, but our proposal is internally consistent and is testable. 

I have improved the description of the model of the fatty acid vesicle mentioning how this gives rise to concentration increase, beginning on line 397 of the revised version. The section describing the homochirality has now also been amplified and the redaction improved, beginning on line 576 of the new version. For more details on these phenomena, please see our papers [3,4,5]. 

[1] Hardy, J. T.: The sea-surface Microlayer: Biology, Chemistry and Anthropogenic Enrichment, Prog. Oceanogr., 11, 307–328, 1982. 

[2] Grammatika, M. and Zimmerman, W. B.: Microhydrodynamics of flotation processes in the sea-surface layer, Dynam. Atmos. Oceans, 34, 327–348, 2001. 

[3] Michaelian, K. The Dissipative Photochemical Origin of Life: UVC Abiogenesis of Adenine. Entropy 2021, 23, 217. https://doi.org/10.3390/e23020217 

[4] Michaelian, K. Homochirality through Photon-Induced Denaturing of RNA/DNA at the Origin of Life. Life 2018, 8, 21. https://doi.org/10.3390/life8020021 

[5] Karo Michaelian, Norberto Santillán Padilla, UVC photon-induced denaturing of DNA: A possible dissipative route to Archean enzyme-less replication, Heliyon, Volume 5, Issue 6, 2019, e01902, ISSN 2405-8440, https://doi.org/10.1016/j.heliyon.2019.e01902 

[6] Mejía Morales, J.; Michaelian, K. Photon Dissipation as the Origin of Information Encoding in RNA and DNA. Entropy 2020, 22, 940. https://doi.org/10.3390/e22090940 

Reviewer 3 Report

The article by Karo Michaelian presents a detailed theory on pre-biotic and biotic evolution based on non-equilibrium thermodynamics. It focuses on photon dissipation as the main physical phenomenon that drives the formation and selection of prebiotic molecules as well as the evolution of organic life. According to this theory, all systems are being selected for their ability to dissipate photonic energy. It accounts for the UV light spectrum of the early Earth and even includes the recent problem of extensive carbon dioxide emissions.

Generally, the argumentation is clear and concise, with well-founded treatments based on classical irreversible thermodynamics. However, I see a clear problem with the restriction on photon dissipation as the sole physical basis. I believe that the driving force for pre-biotic evolution (and of course biological evolution as well) is much more than only the dissipation of photonic energy. Strikingly, the author does not mention catalysis respectively the enzymatic function of biomolecules even once! How can we talk about evolution or the origin of life without including catalytic interactions? How should any metabolic cycle form under these circumstances? What about biomacromolecules such as proteins or polysaccharides? What about membrane structures, the formation of ribosomes, energy metabolism on something like an early ATPase?  And cannot the formation of molecular fragments contribute to increasing entropy as well?

When dealing with Darwinian evolution of organisms, this problem becomes even more virulent. We know that microorganisms evolve in the depth of the Erath’s crust, in absence of any radiation. And: the conclusion that life is carbon based due to the formation of light-absorbing conjugations is simply adventurous. Overall, as much as I like the way that evolution is being treated in context with entropy, this manuscript has a very serious flaw: It is drastically reducing life to a single physical phenomenon in a way that, I am afraid, only very few members of the scientific community are willing to accept.

I see only one possible correction that would make this paper acceptable: a significant reduction of the claim. If the overall message would be to offer a thermodynamic treatment of a specific aspect of evolution, the development of UV-absorbing compounds, then I could recommend the acceptance of the given manuscript.

Author Response

I thank the reviewer for their kind words and for their comments that have helped improve the manuscript. The reviewer's comments and queries are in bold face in the following.  

Generally, the argumentation is clear and concise, with well founded treatments based on classical irreversible thermodynamics. However, I see a clear problem with the restriction on photon dissipation as the sole physical basis. I believe that the driving force for pre-biotic evolution (and of course biological evolution as well) is much more than only the dissipation of photonic energy. Strikingly, the author does not mention catalysis respectively the enzymatic function of biomolecules even once! How can we talk about evolution or the origin of life without including catalytic interactions? How should any metabolic cycle form under these circumstances?  

Catalysis involving biomolecules, or biosystems in general, was, in fact, mentioned 12 times in the original version of the manuscript. Catalysis was mentioned in the production of adenine (DAMN acts as a catalyst for its own production, see [1] for details), in the template directed (autocatalytic) replication of RNA and DNA, in the catalytic template extension of single strand oligos using Mg+2 or Fe+2 ions, in the association of aromatic amino acids (antenna molecules) with nucleic acid which foments replication (Fig. 9), in the water cycle catalyzed by the photon dissipating pigments in plants giving rise to a greener more photon dissipating planet, animals catalyzing the spread of nutrients and seeds for the plants, for example, the top predators (wolves) keeping the elk on the move catalyze the greening of Yellowstone National park.  

It is, in fact, the auto- and cross-catalysis that makes our system non-linear, and it is this which gives the system multiple stationary states [1]. Most importantly, I mentioned catalysis as being the reason for evolutionary stochastic selection of higher peaks on the entropy production surface because catalytic and autocatalytic processes usually have a larger catchment basin in a generalized phase space (see Fig. 7 of manuscript). This, of course, applies to all metabolic cycles. 

What about biomacromolecules such as proteins or polysaccharides? What about membrane structures, the formation of ribosomes, energy metabolism on something like an early ATPase?  

There are, of course, a very large number of events that occurred in the origin and evolution of life. In the manuscript I have attempted to consider some of the most representative of the different kinds of dissipative structuring involving thermodynamic selection which have occurred over life’s history. Examples given in the manuscript were; the dissipative structuring of the fundamental molecules, their association or complexation, the thermodynamic utility of lipid membranes, symbiosis, predator-prey relation, biotic-abiotic coupling, and biosphere evolution. In papers published elsewhere, we have treated, in detail, from the same perspective, these and other important events; for example, information encoding (stereochemical era) [2], vesicle formation [3], homochirality [4], biological catalysis of the water cycle [5]. A saccharide paper is in the works. We believe that photon dissipation was the most important energy metabolism at the origin of life and it is still, by far, the greatest energy metabolism occurring in life today. 

And cannot the formation of molecular fragments contribute to increasing entropy as well?  

The configurational entropy production of molecular fragmentation is very small in comparison to the entropy production of photon dissipation in the region of photon dissipative structuring (part “b” of Fig. 7). This would be more important in the regime of short UV-C wavelengths which cannot lead to dissipative structuring but instead cause fragmentation. This was mentioned in the description of part “a” in the caption of figure 7. 

When dealing with Darwinian evolution of organisms, this problem becomes even more virulent. We know that microorganisms evolve in the depth of the Erath’s crust, in absence of any radiation. And: the conclusion that life is carbon based due to the formation of light-absorbing conjugations is simply adventurous. Overall, as much as I like the way that evolution is being treated in context with entropy, this manuscript has a very serious flaw: It is drastically reducing life to a single physical phenomenon in a way that, I am afraid, only very few members of the scientific community are willing to accept.  

In section 3.1 of the original manuscript, beginning on line 268, I listed the empirical evidence for a UV-C light-based origin of life. I have now included an augmented list of these beginning on line 277 in the new version of the manuscript. These include; 1) the free energy available in UV light of wavelength less than 300 nm arriving at Earth's surface was most likely more than 1000 times that of all other energy sources combined [6], 2) the maximum in the strong absorption spectrum of many of these molecules coincides with the predicted window in the Archean atmosphere (Figure 2), 3) many of the fundamental molecules of life are endowed with peaked conical intersections giving them a broad absorption band and high quantum yield for rapid (picosecond) dissipation of the photon-induced electronic excitation energy into vibrational energy of molecular atomic coordinates, and finally into the surrounding water solvent (Fig. 4), 4) many photochemical routes to the synthesis of nucleic acids, amino acids, fatty acids, cofactors and metabolites, sugars, and other pigments from common precursor molecules have been identified at these wavelengths, 5) the rate of photon dissipation within the Archean window generally increases after each incremental step on route to synthesis, a behavior strongly suggestive of dissipative structuring [7], 6) even minor transmutations (e.g., tautomerizations or methylations) of the fundamental molecules, which often, in fact, endows them with lower Gibb’s free energy, eliminates or significantly reduces their special photon absorption and dissipation properties. These extraordinary absorption and dissipative characteristics of the fundamental molecules of life in the UV-C would have to be considered as mere accidents or curiosities if UV-C light and light absorbing carbon conjugations had nothing to do with the origin of life.  

On line 756 (line 495 and 829 new version) I mentioned that not all entropy production is related to photon dissipation. However, photon dissipation is by far the greatest entropy production process in the biosphere today. This starts with the dissipation of photons in organic pigments and the coupling of this heat of dissipation to the water cycle. The dissipation of chemical potential is only a very tiny, and secondary, fraction (occurring, overwhelmingly, on material first formed through photosynthesis, which couples directly to photon dissipation). For example, plant and cyanobacteria make up 99.5% by mass of all the biomass on Earth [8]. Animals make up less than 0.5% of the biomass [8] and their greatest contribution to the entropy production of the biosphere is not the dissipation of chemical potential, but rather their acting as catalysts to spread of nutrients and seeds over the surface of Earth for the plants on land and the cyanobacteria at the ocean surface, which are the great photon dissipators. 

In summary, light is, and has always been, the primary generalized chemical potential driving life. Other generalized chemical potentials are sporadic, localized, or secondary.  

I see only one possible correction that would make this paper acceptable: a significant reduction of the claim. If the overall message would be to offer a thermodynamic treatment of a specific aspect of evolution, the development of UV-absorbing compounds, then I could recommend the acceptance of the given manuscript. 

The theme of this manuscript, as the title and abstract make clear, is to provide a non-equilibrium thermodynamic foundation, derived from physical law, for describing material interaction with its environment at all scales. A “natural thermodynamic selection” of characteristics of structures (or processes), based stochastically on increases in the global rate of dissipation of the prevailing solar spectrum was described. This implied providing a detailed mechanistic description of such a thermodynamic selection at different biotic-abiotic levels, from the beginning of molecular life in the Archean to the biosphere of today. I believe that I have accomplished this in the manuscript. We have already published several papers on the “development of UV-absorbing compounds” [9,10,11]. This is not the theme of the present paper. 

[1] Michaelian, K. The Dissipative Photochemical Origin of Life: UVC Abiogenesis of Adenine. Entropy 2021, 23, 217. https://doi.org/10.3390/e23020217 

[2] Mejía Morales, J.; Michaelian, K. Photon Dissipation as the Origin of Information Encoding in RNA and DNA. Entropy 2020, 22, 940. https://doi.org/10.3390/e22090940 

[3] Michaelian, K. and Rodríguez, O. (2019). Prebiotic fatty acid vesicles through photochemical dissipative structuring. Revista Cubana de Química, 31 (3), 354-370. 

[4] Michaelian, K. Homochirality through Photon-Induced Denaturing of RNA/DNA at the Origin of Life. Life 2018, 8, 21. https://doi.org/10.3390/life8020021 

[5] Michaelian, K., Biological catalysis of the hydrological cycle: lifes thermodynamic function, Hydrol. Earth Syst. Sci. 16, 2629–2645, 2012, https://hess.copernicus.org/articles/16/2629/2012/. 

[6] Miller, S.L.; Orgel, L., The origins of life on the earth; Prentice-Hall, Englewood Cliffs, N.J., 1974. 

[7] Karo Michaelian, Microscopic dissipative structuring and proliferation at the origin of life, Heliyon, Volume 3, Issue 10, 2017, e00424, ISSN 2405-8440, https://doi.org/10.1016/j.heliyon.2017.e00424. 

[8] Bar-On, R. Phillips, R. Milo, PNAS, 2018, 115 (25) 6506-6511. 

[9] Michaelian, K., Simeonov, A., Fundamental molecules of life are pigments which arose and co-evolved as a response to the thermodynamic imperative of dissipating the prevailing solar spectrum, Biogeosciences, 2015; 12(16), 4913-4937. 

[10] Michaelian, K., Simeonov, A.,Thermodynamic Explanation for the Cosmic Ubiquity of Organic Pigments. Astrobiol Outreach (2017) 5:156. doi:10.4172/2332-2519.1000156 

[11] Michaelian, K., Cano Mateo, R.E. A Photon Force and Flow for Dissipative Structuring: Application to Pigments, Plants and Ecosystems. Entropy 2022, 24, 76. https://doi.org/10.3390/e240

Reviewer 4 Report

This is an unusual and interesting paper which presents an idea on how chemical evolution in prebiotic times or evolution as a whole could fit into a non-equilibrium thermodynamics setting which involves maximization of the rate of entropy production.

Because the paper includes some interesting proposals, I can recommend its acceptance after the following points have been addressed:

1) In particular, the author proposes that dissipation of photon energy from the solar spectrum at the earth's surface could explain the photochemical synthesis of the molecules of life (exemplified by the formation of adenine from HCN), even when entropy of the products is lower (or Gibbs free energy is increasing for certain steps in the mechanism).

This is an interesting proposal, but I missed that the author at least mentions that the accepted opinion is the non-photochemical spontaneous synthesis of adenine catalyzed by NH3 and H2O, see e.g.:

Proc. Natl. Acad. Sci.  2007, 104 (44), 17272-17277. This work should be cited on page 10.

2) Figure 8 shows not only three peaks (maxima of entropy production) but also two local minima of entropy production. Could the author please also comment on them ?

According to the Glansdorff-Prigogine general evolution criterion, evolution of non-equilibrium steady states (even for non-linear systems) seeks to minimize, rather than maximize, entropy production - which is a consequence of the second law of thermodynamics. However, I agree that the influx of a time-reversal non-invariant form of energy in open systems (e.g., light) might conceivably create a "principle of maximum entropy production" - at least for the biosphere, even though the physics behind this has not been worked out yet.

This point has also been made in the SI section of Phys. Chem. Chem. Phys 2023, 25, 1734-1754.

3) More specifically, the author of the present work claims that biological homochirality has been achieved at a supramolecular level of aggregation/disaggregation of DNA (and RNA) strands. However, the transition from a racemic to a homochiral world would have likely occurred *before* biopolymers became available (because of the constituent enantiomerically pure ribose and desoxyribose molecules employed in them, otherwise the formation of stable secondary structures would be quite unlikely).

This is the topic of the abovementioned PCCP paper, which makes the case for a symmetry breaking emergent behaviour of the formose reaction by which carbohydrates had been generated at the origin of life. This work should therefore be cited, when discussing "homochirality of the nucleic acids" on page 18.

Author Response

I thank the reviewer for their kind words and for their comments that have helped improve the manuscript. The reviewer's comments and queries are in bold face in the following.  

This is an unusual and interesting paper which presents an idea on how chemical evolution in prebiotic times or evolution as a whole could fit into a non-equilibrium thermodynamics setting which involves maximization of the rate of entropy production. 

Because the paper includes some interesting proposals, I can recommend its acceptance after the following points have been addressed: 

1) In particular, the author proposes that dissipation of photon energy from the solar spectrum at the earth's surface could explain the photochemical synthesis of the molecules of life (exemplified by the formation of adenine from HCN), even when entropy of the products is lower (or Gibbs free energy is increasing for certain steps in the mechanism). 

This is an interesting proposal, but I missed that the author at least mentions that the accepted opinion is the non-photochemical spontaneous synthesis of adenine catalyzed by NH3 and H2O, see e.g.: Proc. Natl. Acad. Sci. 2007, 104 (44), 17272-17277. This work should be cited on page 10.  

This work is now cited on p. 11, at the end of the caption of figure 5. 

2) Figure 8 shows not only three peaks (maxima of entropy production) but also two local minima of entropy production. Could the author please also comment on them?  

Minima on the entropy production surface represent unstable stationary states. An analogy would be the peaks on the potential energy surface of a system, representing unstable free energy local maxima. Any small fluctuation will cause the system to leave these points and move away from the extremum (downward on the potential energy surface and upward to higher entropy production on the entropy production surface). It would be very improbable to find nature at these extrema. 

According to the Glansdorff-Prigogine general evolution criterion, evolution of non-equilibrium steady states (even for non-linear systems) seeks to minimize, rather than maximize, entropy production - which is a consequence of the second law of thermodynamics. 

Glansdorff and Prigogine define the total entropy production as a sum of a part due to the variation of the forces d_X P/dt and the other part due to the variation of the flows d_J P/dt, 

dP/dt = d_X P/dt + d_J P/dt . 

In the case of linear systems, d_X P/dt can be shown to be identical to d_J P/dt, and this ensures that the total entropy production dP/dt is negative definite (always decreases). However, for non-linear systems, d_XP/dt and d_JP/dt are not identical, and Glansdorff and Prigogine have proven that only d_X P/dt is negative definite and that d_J P/dt can take on either sign. Therefore, for non-linear systems there does not exist a total differential (potential) that can be optimized to determine the evolution of the system. This corresponds to the fact that for non-linear systems, multiple stationary states exist and therefore many maxima exist on the entropy production surface (e.g., figure 7 of the manuscript). To go from one peak to another, the system must go through a critical point, corresponding to a point on the boundary between stationary state attraction basins (peaks) before arriving at the other stationary state with a different entropy production peak. This can happen through stochastic fluctuation. Scenarios corresponding to autocatalytic processes have larger peaks in entropy production surface and larger attraction basins, which makes them more probable. However, there is no potential, or function, that can be optimized to lead from one peak to the other. The system could also move to a lower entropy production peak. It is just that those states with larger entropy production (e.g., involving catalysis or autocatalysis) have generally larger attraction basins and thus will be more probable under stochastic fluctuation. Summarizing, for non-linear systems, higher entropy production is more probable in systems with positive feedback (autocatalytic) but this is not a general rule or optimizing principle like MEP. 

 However, I agree that the influx of a time-reversal non-invariant form of energy in open systems (e.g., light) might conceivably create a "principle of maximum entropy production" - at least for the biosphere, even though the physics behind this has not been worked out yet. This point has also been made in the SI section of Phys. Chem. Chem. Phys 2023, 25, 1734-1754.  

Rather than call it a “principle of maximum entropy production”, I would prefer to call it a stochastic probability of generally increasing entropy production, with only local maximization of entropy production within a peak (local attractor of a stationary state) in a generalized phase space.  

More specifically, the author of the present work claims that biological homochirality has been achieved at a supramolecular level of aggregation/disaggregation of DNA (and RNA) strands. However, the transition from a racemic to a homochiral world would have likely occurred *before* biopolymers became available (because of the constituent enantiomerically pure ribose and desoxyribose molecules employed in them, otherwise the formation of stable secondary structures would be quite unlikely). This is the topic of the above mentioned PCCP paper, which makes the case for a symmetry breaking emergent behavior of the formose reaction by which carbohydrates had been generated at the origin of life. This work should therefore be cited, when discussing "homochirality of the nucleic acids" on page 18. 

Let us assume that polymerization of racemic nucleotides occurs randomly under UV-C light, forming single strands of nucleic acid of mixed chirality. Some of those initial oligos will, by chance, be of a pure chirality. For example, the probability of a 10 base single strand being of a single (pure) chirality would be 2x(0.5)^10 = 0.00195 which is one in 512, which is not negligible. It will be only these that will replicate faithfully through overnight extension with the aid of magnesium ions since mixed chirality frustrates complete extension. This will eventually lead to two sets of double strands of different pure chirality (i.e., right-handed oligos and left-handed oligos). Those of the chirality corresponding to that of the circular polarization of the submarine light of the afternoon will have a somewhat greater chance of UV-C induced denaturing (Fig. 9) because of the higher surface temperature in the afternoon so will soon come to dominate [1]. Racemization under UV-C light, or temperature, of the single nucleotides, or short oligos, will lead to a constant supply of nucleotides of the correct chirality for incorporation in the predominant homochiral oligos. 

The PCCP work on symmetry breaking emergent behavior of the formose reaction by which carbohydrates form, mentioned by the reviewer, has been considered and cited in the new version of the manuscript (line 588). 

[1] Michaelian, K. Homochirality through Photon-Induced Denaturing of RNA/DNA at the Origin of Life. Life 2018, 8, 21. https://doi.org/10.3390/life8020021 

Round 2

Reviewer 1 Report

I suggest to accept the paper.

Author Response

I thank the reviewer for their work in reviewing the manuscript and for their recommendation to publish the manuscript.

Reviewer 2 Report

ok

Author Response

(The authors gave the same response as above.)

Reviewer 3 Report

Dear Editors, dear author, 

I see improvements in the manuscript, but I still do have problems with the overall claim that all evolution is exclusively driven by UV light absorption. For simplicity, I will refer to the main issues in the text:

Catalysis involving biomolecules, or biosystems in general, was, in fact, mentioned 12 times in the original version of the manuscript. Catalysis was mentioned in the production of adenine (DAMN acts as a catalyst for its own production, see [1] for details), in the template directed (autocatalytic) replication of RNA and DNA, in the catalytic template extension of single strand oligos using Mg+2 or Fe+2 ions, in the association of aromatic amino acids (antenna molecules) with nucleic acid which foments replication (Fig. 9), in the water cycle catalyzed by the photon dissipating pigments in plants giving rise to a greener more photon dissipating planet, animals catalyzing the spread of nutrients and seeds for the plants, for example, the top predators (wolves) keeping the elk on the move catalyze the greening of Yellowstone National park.  

It is, in fact, the auto- and cross-catalysis that makes our system non-linear, and it is this which gives the system multiple stationary states [1]. Most importantly, I mentioned catalysis as being the reason for evolutionary stochastic selection of higher peaks on the entropy production surface because catalytic and autocatalytic processes usually have a larger catchment basin in a generalized phase space (see Fig. 7 of manuscript). This, of course, applies to all metabolic cycles. 

It is not the term catalysis I am missing here, it is the reference to enzymatic catalysis. The first living cell could have only existed with an energy metabolism based on a complex catalytic network. This was based on a large set of enzymes, each one with an optimized amino acid sequence for the given purpose. How did these enzymes develop? How was the more efficient enzyme selected against the less efficient one? Definitely not by its light absorption properties.

There are, of course, a very large number of events that occurred in the origin and evolution of life. In the manuscript I have attempted to consider some of the most representative of the different kinds of dissipative structuring involving thermodynamic selection which have occurred over life’s history. Examples given in the manuscript were; the dissipative structuring of the fundamental molecules, their association or complexation, the thermodynamic utility of lipid membranes, symbiosis, predator-prey relation, biotic-abiotic coupling, and biosphere evolution. In papers published elsewhere, we have treated, in detail, from the same perspective, these and other important events; for example, information encoding (stereochemical era) [2], vesicle formation [3], homochirality [4], biological catalysis of the water cycle [5]. A saccharide paper is in the works. We believe that photon dissipation was the most important energy metabolism at the origin of life and it is still, by far, the greatest energy metabolism occurring in life today. 

I would just wish that this fact would be stated more clearly in the given manuscript: that it deals with a very specific aspect of evolution, the development of UV absorbing species. To my opinion, this should be clearly mentioned in the abstract.

The configurational entropy production of molecular fragmentation is very small in comparison to the entropy production of photon dissipation in the region of photon dissipative structuring (part “b” of Fig. 7). This would be more important in the regime of short UV-C wavelengths which cannot lead to dissipative structuring but instead cause fragmentation. This was mentioned in the description of part “a” in the caption of figure 7. 

The entropy gain by molecular fragmentation may be small, but it nevertheless provides the driving force for most part of natural selection. If you look at driving forces in metabolitic steps, they are always very small. Biochemistry even seems to avoid all strongly exergonic reactions. 

 In section 3.1 of the original manuscript, beginning on line 268, I listed the empirical evidence for a UV-C light-based origin of life. I have now included an augmented list of these beginning on line 277 in the new version of the manuscript. These include; 1) the free energy available in UV light of wavelength less than 300 nm arriving at Earth's surface was most likely more than 1000 times that of all other energy sources combined [6], 2) the maximum in the strong absorption spectrum of many of these molecules coincides with the predicted window in the Archean atmosphere (Figure 2), 3) many of the fundamental molecules of life are endowed with peaked conical intersections giving them a broad absorption band and high quantum yield for rapid (picosecond) dissipation of the photon-induced electronic excitation energy into vibrational energy of molecular atomic coordinates, and finally into the surrounding water solvent (Fig. 4), 4) many photochemical routes to the synthesis of nucleic acids, amino acids, fatty acids, cofactors and metabolites, sugars, and other pigments from common precursor molecules have been identified at these wavelengths, 5) the rate of photon dissipation within the Archean window generally increases after each incremental step on route to synthesis, a behavior strongly suggestive of dissipative structuring [7], 6) even minor transmutations (e.g., tautomerizations or methylations) of the fundamental molecules, which often, in fact, endows them with lower Gibb’s free energy, eliminates or significantly reduces their special photon absorption and dissipation properties. These extraordinary absorption and dissipative characteristics of the fundamental molecules of life in the UV-C would have to be considered as mere accidents or curiosities if UV-C light and light absorbing carbon conjugations had nothing to do with the origin of life.  

Still, I want to come back to the issue that microorganisms have evolved in the depth of the Earth's crust. In this environment, a very efficient evolution has taken place in absence of all light sources. This allows for the conclusion that general evolution of organisms can not be driven by light absorption. I agree, however, that the described mechanism may hold for particular molecular structures or metabolitic steps for those organisms which live at the surface.

On line 756 (line 495 and 829 new version) I mentioned that not all entropy production is related to photon dissipation. However, photon dissipation is by far the greatest entropy production process in the biosphere today. This starts with the dissipation of photons in organic pigments and the coupling of this heat of dissipation to the water cycle. The dissipation of chemical potential is only a very tiny, and secondary, fraction (occurring, overwhelmingly, on material first formed through photosynthesis, which couples directly to photon dissipation). For example, plant and cyanobacteria make up 99.5% by mass of all the biomass on Earth [8]. Animals make up less than 0.5% of the biomass [8] and their greatest contribution to the entropy production of the biosphere is not the dissipation of chemical potential, but rather their acting as catalysts to spread of nutrients and seeds over the surface of Earth for the plants on land and the cyanobacteria at the ocean surface, which are the great photon dissipators. 

In summary, light is, and has always been, the primary generalized chemical potential driving life. Other generalized chemical potentials are sporadic, localized, or secondary.  

 Again, why should the amount of entropy gain be a relevant argument? It would rather be plausible that early life avoided large energy dissipation, as such a process very often proves to be destructive. All recent biochemical cycles (like, e.g. the Krebs cycle or the related oxydative pathways) are characterized by minimized steps of entropy production. 

The theme of this manuscript, as the title and abstract make clear, is to provide a non-equilibrium thermodynamic foundation, derived from physical law, for describing material interaction with its environment at all scales. A “natural thermodynamic selection” of characteristics of structures (or processes), based stochastically on increases in the global rate of dissipation of the prevailing solar spectrum was described. This implied providing a detailed mechanistic description of such a thermodynamic selection at different biotic-abiotic levels, from the beginning of molecular life in the Archean to the biosphere of today. I believe that I have accomplished this in the manuscript. We have already published several papers on the “development of UV-absorbing compounds” [9,10,11]. This is not the theme of the present paper. 

As mentioned before, I very much like the approach to use non-equilibrium thermodynamics to understand the nature of evolution. Energy dissipation of radiation energy may be a good example to demonstrate this idea. However, I believe that it is not appropriate to assign evolution exclusively to light absorption phenomena. The manuscript should account to the fact that there actually is evolution in absence of solar radiation. 

Author Response

I thank the reviewer for their comments and suggestions on the revised version of the manuscript. I have now included a new section (section 5.3) to address specifically the reviewers concern of the evolution of chemoautotrophic organisms. In the following, the reviewer's comments are in boldface, reorganized into themes, and my response. 

The first living cell could have only existed with an energy metabolism based on a complex catalytic network. This was based on a large set of enzymes, each one with an optimized amino acid sequence for the given purpose. How did these enzymes develop? How was the more efficient enzyme selected against the less efficient one? Definitely not by its light absorption properties. 

Still, I want to come back to the issue that microorganisms have evolved in the depth of the Earth's crust. In this environment, a very efficient evolution has taken place in absence of all light sources. This allows for the conclusion that general evolution of organisms can not be driven by light absorption. I agree, however, that the described mechanism may hold for particular molecular structures or metabolitic steps for those organisms which live at the surface. 

I first emphasize the distinction between the perspective presented in the manuscript and the traditional perspective. In the traditional framework, organisms are imagined to have somehow arisen and then struggle to “optimize” their enzymatic metabolic pathways to improve their own differential reproductive success. We criticize this perspective throughout the text because it has no physical foundation and leads to problems and paradoxes. Instead, our perspective is that of non-equilibrium thermodynamic theory where generalized flows, i.e., dissipative structures, arise “spontaneously” in response to applied generalized forces (gradients of potentials, e.g., the imposed photon potential, chemical potentials, electric potentials, etc.) for no reason other than the thermodynamic one to increase the dissipation of the imposed potential (an abiotic example would be the “spontaneous” origin of convection cells, or a hurricane, to dissipate a temperature gradient). There is no optimization other than this stochastic thermodynamic one relevant to any irreversible process.  

In direct answer to the reviewer, the following new section 5.3 entitled “Evolution of Chemotrophic Organisms as Catalysts for Phototrophic Organisms” was included in the new version of the manuscript (line 649). 

“The solar photon potential drove most of the dissipative structuring known as life. However, today there also exist chemotrophic organisms (e.g., bacteria living at hydrothermal vents and underground, and animals) that dissipate a chemical potential. Most of these chemical potentials are secondary, resulting from processes (e.g., photosynthesis) that dissipate the solar photon potential. For example, most bacteria underground or at the bottom of the oceans are dissipating the chemical potential available from buried organic material or the rain of organic material from the surface, respectively. Animals dissipate the chemical potential that was originally produced in plants and cyanobacteria. 

There also exist, however, chemolithotrophic organisms (bacteria and archaea) at the surface, under water and underground, which survive independently of sunlight. They have special enzymes for oxidation of inorganic electron donors (e.g., hydrogenase, carbon monoxide oxidase, nitrite oxidoreductase, sulfite oxidase, etc). As an example, chemolithotrophic microbes, found kilometers below the ocean and land surfaces, appear to depend exclusively on geologically produced hydrogen and sulfur compounds. Hydrogen comes from the decomposition of methane by high temperatures and pressures, or the decomposition of water by the radioactive decay of uranium, thorium and potassium. All of these microbes have been shown to be genetically related to cousins on the surface, having been buried and separated from them for several million to tens of millions of years. The metabolism of these ocean or land buried organisms is 10,000 to one million times slower than at the surface. The evolution of these organisms is, therefore, also correspondingly much slower than at the surface, being affected only by point mutation and genetic drift \cite{OrsiEtAl2021}.  

There is no evidence that chemotrophic organisms preceded phototrophic organisms. In fact, a large set of optimized amino acid sequences (enzymes) would have been needed for a primordial chemoautotrophic protocell (as the reviewer observes) because the chemical free energy available in inorganic substances (redox potential) is orders of magnitude smaller than the free energy available in a single UV-C photon, thus inhibiting reactions involving carbon covalent bonding which commonly have high activation barriers, unless such enzymes were already available at the origin of life. For this reason, the first “living” cell most likely arose as a dissipative structure under the soft UV-C light potential rather than under a chemical potential. Evidence for this was presented in the extraordinary UV-C photochemical properties found for the fundamental molecules of life listed in section \ref{sec:AbsLight}. 

The chemotrophic organisms, although unlikely primordial, are not irrelevant to photon dissipation. Through their metabolism, excrement, and death, they provide a nutrient recycling service to the phototrophic organisms. In our thermodynamic view, they are thus, first and foremost, catalysts for the phototrophic organisms dissipating photons. 

The enzymes of these chemotrophic organisms were selected over millions of years of evolution almost exclusively according to how they improve their catalytic effectiveness in aiding photon dissipation. The mechanism of this thermodynamic selection of enzymes leading to increased global photon dissipation, is symbiosis (cross catalysis). As an example, the greater the ability of underground rhizobacteria bacteria to provide the required nutrients to the plants (e.g., nitrogen) the greater the number of chlorophyll and other pigment molecules and thus the greater the photon dissipation rate in the leaves of the plants and therefore the greater the osmotic pressure to draw up water with nutrients from the roots, allowing the symbiotic bacteria to acquire chemical potential and photosynthetic carbohydrates from plants for their own maintenance. Gene selection for both the plant and the chemotrophic bacteria is thus optimized for global photon dissipation, not for the differential reproductive success of either one of the symbionts. Growth, the increase in numbers of phototrophic organisms and their spread over the whole of Earth’s surface (traditional evolutionary perspective), is just one of many ways that global photon dissipation is increased. Others include, 1) increases in the region of wavelength absorption of plant pigments to cover the whole solar spectrum, 2) increasing the rate of photon-induced excited state dissipation to the ground state, allowing the pigment to process (dissipate) more photons, 3) coupling of the dissipation of photons in the leaves to the water cycle and other abiotic processes (shifting the wavelength of the outgoing photons even further towards the infrared), 4) exudation of pigments into the environment (e.g., oxygen, mycosporine-like amino acids, isoprenes, etc.). All of these factors important to photon dissipation are “optimized” through the natural thermodynamic selection of enzymes of both photoautotrophic and chemotrophic organisms.” 

I have also included the following paragraph in the Conclusions; 

“Life is not exclusively ordered through dissipation of the solar photon potential, but this potential is by far the most important, all others being either obtained secondarily from the photon potential or very rare (e.g., chemolithotrophy). The evolutionary process of the complexation of life would thus most likely have gone through the following major stages of dissipative structuring; 1) UV-C pigments (fundamental molecules), 2) photon dissipative molecular complexes, 3) protocells consisting of UV-C dissipative molecular complexes within vesicles, 4) exudation into the environment of the pigments oxygen, mycosporine-like amino acids and isoprenes, to dissipate UV light, shielding the more complex but delicate enzymes (e.g., van der Waals and hydrogen bonded folded proteins), allowing the dissipative structuring using, and the dissipation of, visible photons (photosynthesis), the Krebs cycle, as well as chemolithotrophy, 5) symbiosis, parasitism and mutualism, 6) coupling of biotic and abiotic dissipative processes, 7) intelligence and technology to spread plant nutrients (e.g. phosphorus and CO$_2$), to dissipate underground stored chemical potential (e.g., coal and petroleum), and to bootstrap carbon-based photon dissipation on other planets.” 

I would just wish that this fact would be stated more clearly in the given manuscript: that it deals with a very specific aspect of evolution, the development of UV absorbing species. To my opinion, this should be clearly mentioned in the abstract. 

But the manuscript not only deals with the development of UV absorbing species. It deals with carbon-based material response in general to the application of an external photon potential. Different examples with their respective mechanisms of thermodynamic selection were chosen from abiotic-biotic systems at different biotic-abiotic levels, from the UV absorbing pigments at the origin of life to the visible absorbing biosphere of today. The mechanism of thermodynamic selection of chemotrophic organisms is now also included in a new section 5.3. 

The entropy gain by molecular fragmentation may be small, but it nevertheless provides the driving force for most part of natural selection. If you look at driving forces in metabolitic steps, they are always very small. Biochemistry even seems to avoid all strongly exergonic reactions. 

Again, why should the amount of entropy gain be a relevant argument? It would rather be plausible that early life avoided large energy dissipation, as such a process very often proves to be destructive. All recent biochemical cycles (like, e.g. the Krebs cycle or the related oxydative pathways) are characterized by minimized steps of entropy production. 

Catabolisis is important in the evolution of chemotrophic organisms, but I would say that the driving force for natural selection leading to evolution, as emphasize in the manuscript, is not molecular fragmentation, but rather photon dissipation leading to molecular production (dissipative structuring). It is these dissipatively structured (photosynthetic) molecules that are the source molecules for catabolism. 

Dissipative structuring is part of non-equilibrium thermodynamic theory, and describes how material responds to the application of an external potential. Dissipative structuring describes the dynamics of all macroscopic process, whatsoever, without exception, abiotic or biotic. These processes arise “spontaneously” to dissipate the imposed thermodynamic potential (i.e., entropy production, or in the words of the reviewer “entropy gain”). Processes do not arise because of some kind of mysterious inherent “vitality” of material, instead, they arise as a generalized flow (heat flow, material flow, chemical reaction rate, photochemical dissipation, etc.) created by a respective generalized force (gradient of temperature, gradient of concentration, chemical affinity, photon force [1], etc.), and these flows arise to dissipate the potential (the force is the gradient of the potential). Dissipation is the agent of order, the greater the dissipation, normally the greater the order. For a fixed force over the system, entropy production (dissipation) is a measure of the strength of the created flow (dissipative structuring) since entropy production is just flow J times force X, i.e., d_i S/dt = J*X. The entropy produced does not accumulate within the flow (the organism) but is exported to the environment (the infrared light given off by the biosphere exported to outer space). Evolution is the process of generally increasing entropy production over time. 

The material of the system reduces its entropy (the “structuring” of “dissipative structuring”), but that of the environment is greatly increased (the “dissipation” of “dissipative structuring”). Dissipation is the agent of organization.  

The Krebs cycle is powered by the secondary chemical potential of organic molecules obtained from visible photon dissipation in photosynthesis. It is one of many coupled processes composing the biosphere. It may be that the Krebs cycle is linear (or a least our modeling of it is linear) and thus appears to be minimizing entropy production (if, indeed, it could be isolated). However, it is not correct to draw conclusions from this concerning the importance of photon dissipation or entropy production to the entire biosphere. For example, animals are dissipating little but their function as catalysts for the plants and cyanobacteria, which are doing the heavy dissipation, cannot be ignored. If we could remove all the animals from Earth, plants would certainly die off rapidly and the photon dissipation will be dramatically reduced. An example of a real situation given in the manuscript is that of the greening of Yellowstone Park when wolves were re-introduced. 

It seems that chemotrophic bacteria and animals, because they dissipate chemical potential produced by plants and cyanobacteria, tend to minimize their entropy production to maximize the free energy for their work as catalysts to the plants and cyanobacteria while consuming as little plant and cyanobacteria material as possible so as not to significantly reduce the thermodynamic function of cyanobacteria and plants, which is heavy photon dissipation. 

[1] Michaelian, K., Cano Mateo, R.E. A Photon Force and Flow for Dissipative Structuring: Application to Pigments, Plants and Ecosystems. Entropy 2022, 24, 76. https://doi.org/10.3390/e240 

As mentioned before, I very much like the approach to use non-equilibrium thermodynamics to understand the nature of evolution. Energy dissipation of radiation energy may be a good example to demonstrate this idea. However, I believe that it is not appropriate to assign evolution exclusively to light absorption phenomena. The manuscript should account to the fact that there actually is evolution in absence of solar radiation. 

I do not assign evolution exclusively to photon dissipation, but almost exclusively. I have now included a new section (5.3) dedicated to the evolution of chemotrophic organisms, beginning on line 649. 

Round 3

Reviewer 3 Report

With these changes and additions, the author at least opens a minor pathway to the acceptance of other mechanisms of evolution besides the one which is presented in this paper. 

With this, I regard the manuscript to be acceptable for publication in Entropy

Still, I would consider it to be a relatively exotic view on evolution and the origin of life.